

# Simulation of long-term spatiotemporal variations in regional-scale groundwater recharge: Contributions of a water budget approach in southern Quebec

Emmanuel Dubois[1], Marie Larocque[1,2], Sylvain Gagné[1], Guillaume Meyzonnat[1]

[1]Université du Québec à Montréal, Département des sciences de la Terre et de l'atmosphère and GEOTOP Research Center, Pavillon Président-Kennedy, local PK-6151 C.P. 8888, Succursale Centre-Ville, Montréal (Quebec) H3C 3P8 Canada
[2] GRIL Research Center, Département de sciences biologiques, Université de Montréal, Campus MIL, C.P. 6128, Succ. Centre-ville, Montréal (Quebec) H3C 3J7 Canada

*Correspondence to*: Emmanuel Dubois (dubois.emmanuel@courrier.uqam.ca)

**Abstract.** Groundwater recharge (GWR) is recognized to be a strategic hydrologic variable, necessary to estimate when implementing sustainable groundwater management, especially within a global change context. However, its simulation at the regional scale and for long-term conditions is challenging, especially due to the limited availability of spatially-distributed calibration data and to the rather short observed time series. The use of a superficial water budget model to estimate recharge

is appropriate for this task. A reliable regional-scale estimate of GWR that can be updated relatively easily using widely-available data is essential for the implementation of long-term water use policies and is clearly lacking in southern Quebec (Canada; 36 000 km²). This study aims to test the ability of a spatially-distributed water budget model, automatically calibrated with river flow rates and baseflow estimates, to simulate GWR at a regional-scale from 1961 to 2017 in southern Quebec (monthly time step, 500 m x 500 m spatial resolution). The novelty of this work lies in the simulation of the first regional-scale

GWR estimate for southern Quebec and in the development of a robust approach to implement a superficial water budget model at the regional-scale and for a long period. The HydroBudget model was specifically developed by a team at Université du Québec à Montréal for regional-scale simulation and cold climate conditions, and uses parsimonious input data (distributed precipitation, temperature, and runoff curve numbers). The model was regionally calibrated with river flows and baseflows (recursive filter on river flow data), and the automatic calibration procedure of the R package *caRamel* allowed a satisfying

calibration quality (KGE = 0.72) to be reached. Across the study area and based on the exceptionally long spatialized time series, the simulated water budget was divided into 41% runoff (444 mm/yr), 47% actual evapotranspiration (501 mm/yr), and 12% potential groundwater recharge (139 mm/yr). This partitioning was influenced by precipitation, temperature, soil texture, land cover, and topography. Groundwater recharge peaked during spring (44% of annual recharge) and winter (32% of annual recharge). A novel and particularly useful result from this work was to show that the seasonality of recharge was driven by the

regional temperature gradient, with decreasing temperatures from west to east, and that winter GWR presented a statistically significant increasing trend since 1961 due to increased precipitation and warming temperatures. Another original contribution



of this work was to show that at the regional scale, water budget models, such as HydroBudget, can be easily calibrated with river flow measurements and baseflows, and therefore represent a good option with which to acquire knowledge about regional hydrological dynamics. Being accessible, they are a useful approach for scientists, modellers, and stakeholders alike to
understand regional-scale groundwater renewal rates, especially if they can be easily adapted to specific study needs and environments.

## 1 Introduction

Groundwater recharge (GWR) generally refers to the portion of precipitation that infiltrates into the ground and eventually
reaches the water table (Doble and Crosbie, 2017; Scanlon et al., 2002), linking meteorological events to subsequent aquifer responses and defining aquifer renewal rates. GWR is influenced by climate, land cover, vegetation, topography, as well as soil type (Douville et al., 2013; Fu et al., 2019; Jasechko et al., 2014). It is recognized as one of the most strategic hydrologic variables to be estimated for sustainable groundwater management (Foster and Ait-Kadi, 2012; Wada et al., 2010; Zhou, 2009). This is especially true in a global change context, where climate, land cover, and water use are evolving (Green et al., 2011;
Kløve et al., 2014). Consequently, spatiotemporal GWR estimates at the regional scale are particularly important for water resource managers (Ashaolu et al., 2020; Brunner et al., 2004; Larocque et al., 2018).

Models are among the most common ways to estimate GWR, and are usually developed for specific climate and hydrogeological conditions (Healy and Scanlon, 2010; Scanlon et al., 2002). These models vary widely in complexity, from few (< 10) to numerous (> 100) parameters, and from simple input data, such as precipitation and temperature, to detailed
input data (spatialized climate observations, groundwater levels, river flow, soil water content, remote sensing data, etc.), as well as in spatiotemporal resolutions (Abdollahi et al., 2017; Brunner et al., 2004; Chung et al., 2010; Crosbie et al., 2015; Döll and Fiedler, 2008; Hu et al., 2019; Portoghese et al., 2005). GWR assessment based on water budget approaches are widely used, providing useful results and insights on GWR dynamics for an array of local (field scale) to regional studies (several 1,000 km$^2$), and for a wide range of climatic conditions (Abdollahi et al., 2017; Ashaolu et al., 2020; Beigi and Tsai,
2014; Croteau et al., 2010; Dripps and Bradbury, 2007; Dyer, 2019; Portoghese et al., 2005; Zomlot et al., 2015). Nevertheless, water budget models have strong limitations. In such models, GWR is dependent on other terms of the water budget, namely evapotranspiration and runoff (Jasechko et al., 2014), which can be highly uncertain, thus making error propagation significant, especially in arid to semi-arid areas (Crosbie et al., 2015; Scanlon et al., 2002). However, water budget approaches can be appropriate in humid and cold conditions, where substantial amounts of water are stored in the snowpack and then quickly
become available for infiltration during the thaw period. For example, Dripps and Bradbury (2007) acknowledge the lack of representativeness of the daily results of the Soil-Water Balance model (SWB; Westenbroek et al., 2010) while showing that the monthly and annual results for a site in Wisconsin (U.S.A – humid continental climate) are comparable to results obtained from a 2D analytic element groundwater flow model (GFLOW, Hunt et al., 1998). Guay et al. (2013) reached a similar





conclusion, finding similar annual GWR estimates and matching patterns of seasonal variations from a fully distributed model

(CATHY, Comporese et al., 2010) as from a water budget model (HELP, Schroeder et al., 1994) at the field scale (0.2 km$^2$) in Quebec (Canada – humid continental climate).

In southern Quebec (Canada), GWR has been estimated in various studies (Carrier et al., 2013; Chemingui et al., 2015; Croteau et al., 2010; Gagné et al., 2018; Guay et al., 2013; Larocque and Pharand, 2010; Larocque et al., 2013, 2015b, 2018; Lavigne et al., 2010; Lefebvre et al., 2015; Levison et al., 2014, 2016; McCormack and Therrien, 2014; Nastev et al., 2008; Rivard et

al., 2009, 2014; Saby et al., 2016; Talbot-Poutin et al., 2013). Interannual GWR rates estimated from these studies range from 50 mm/yr to more than 450 mm/yr throughout southern Quebec. The studies generally agree in identifying preferential GWR areas (mostly dependent on the nature and the architecture of the unconsolidated quaternary deposits) and periods (spring and fall, linked to snowmelt and to low evapotranspiration rates). However, for a given area, GWR rates and temporal trends can be highly variable from one study to another. This may be due to the different periods of observation (baseflows, groundwater

levels) used as a proxy for GWR or as calibration data for GWR simulations (Rivard et al., 2009), or to the different spatial scales involved. The use of different models can also explain the different results obtained under similar conditions (Larocque et al., 2019; Velázquez et al., 2013).

Establishing an overview of groundwater renewal rates for water resource management and land management based on these data can be quite challenging, especially for non-experts. This is especially true when computational times are long and the

pre-treatment of the calibration data necessary to produce reliable results is burdensome. There is clearly a need for reliable regional-scale estimates of GWR that can be updated relatively easily using data widely available for southern Quebec. This is essential for the implementation of long-term water use policies aimed at adapting to climate and land use change. Considering the spatial scale involved (tens of thousands of km$^2$), the meteorological and river flow rate data available since the 1960s, the lack of groundwater level measurements covering that time span, and the cold and humid climate conditions, a

spatialized transient-state water budget that considers water storage in the snowpack and soil frost appears to be the most appropriate approach for simulating GWR for cold and humid regions (De Vries and Simmer, 2002; Scanlon et al., 2002). This study aims to test the ability of a regional-scale spatially-distributed water budget model, automatically calibrated with river flow rates and baseflow estimates, to simulate GWR over several decades in southern Quebec. The novelty of this work lies in the development of the first regional-scale GWR estimate for southern Quebec and in the development of a robust approach

to implement a superficial water budget model at the regional-scale and for a long period.

HydroBudget (HB) is a parsimonious water budget model (Dubois et al., 2021; Larocque et al. 2013, 2015a, 2015b), capable of handling large amounts of data to model spatially-distributed recharge. It was used here to simulate GWR from 1961 to 2017 for watersheds of the St. Lawrence River in southern Quebec (Canada; 36 000 km$^2$). The study area is first presented, followed by a brief description of the model. The results for southern Quebec are described and discussed in terms of regional

and temporal dynamics, and the contributions to existing knowledge and limitations of the approach are discussed.





## 2 Study area

The study area is located in the province of Quebec (humid continental climate), between the St. Lawrence River and the Canada-USA border, and between the Quebec-Ontario border and Quebec City (35 800 km²) (Fig. 1). It is comprised of the watersheds of eight main tributaries of the St. Lawrence River (numbered 1 to 8 from west to east, and with areas ranging from
460 km² to more than 9 000 km²). Watersheds W1 (Châteaugay River), W2 (Richelieu River), and W4 (Saint-François River) are partially located in the USA (42%, 83%, and 15% of their total areas respectively).

Topography varies between 0 m above sea level (masl) and 1,100 masl (Table 1), with flat, low elevation areas close to the St. Lawrence River (W1, W2, W3, and W7) and higher elevations in the Appalachian mountain range (W4, W5, W6, and W8) associated with steeper slopes. Land cover includes agriculture (ranging from 23% in W8 to 62% in W1), forest (ranging from
26% in W1 to 66% in W8), wetlands (ranging from 3% in W2 and W3, to 14% in W7), urban uses (ranging from 3% in W5, W6, W7, and W8, to 11% in W2), and surface water (ranging from 1% in W1, W3, W5, W6, and W8, to 4% in W4) (Fig. 2a). Agriculture dominates in the watersheds located in the St. Lawrence Platform (W1, W2, and W3), while forest occupies most of the Appalachian watersheds (W4, W6, and W8). Wetlands are more abundant in W7 (Table 1).

Bergeron (2016) used 251 climate stations located in the study area to produce spatially-interpolated temperature and
precipitation data with a daily time step for the 1961-2017 period on a 10 km-resolution grid. The high density of measurements during this period generated minimal error on the interpolated data, with a root mean square error (RMSE) of 3 mm/d for precipitation, 2.5°C for minimal temperature, and 1.5°C for maximal temperature (Bergeron, 2016). Based on these data, average annual precipitation for the study area is 1 090 mm (Table 1), including 275 mm falling as snow during winter (November to March). Precipitation is distributed relatively evenly throughout the year, and although total precipitation is
similar for all of the watersheds, it is higher in the mountainous zones than in the lowlands (Fig. 2c). The average annual temperature is 5°C (Table 1), with average monthly temperature varying between -15°C (in January and February) and 20°C (in July and August). Regionally, temperatures decrease from the southwest to the northeast (Fig. 2d).

The study area includes two main geological units: the sedimentary basin of the St. Lawrence Platform and the Appalachian mountain range (metasedimentary) (Malo, 2018; Fig. 2c). The bedrock of the St. Lawrence Platform is composed of slightly
deformed Combro-Ordovician fractured bedrock consisting of limestone, shale, and mixed shale and fine sandstone. The bedrock of the Appalachians is composed of highly deformed schist, quartzite, and phyllades (Thériault and Malo, 2018).

The bedrock is unevenly covered with unconsolidated Quaternary sediments from the last glaciation-deglaciation cycle, mainly composed of glacial, marine, fluvial, and lacustrine deposits. In this area, sediment depositional modes determine the geological nature of the superficial materials and the pedology (Fig. 2b). Thin (< 5 m) and relatively coarse superficial deposits
(aeolian sands, till, and various glacial deposits) are found over the bedrock in the uphill areas. These deposits get thicker (5 m to 20 m) in the valleys and in topographic depressions, with a wider mix of grain sizes resulting from the different depositional modes (fluvial, lacustrine, and marine). Closer to the St. Lawrence River, the thickness of the Quaternary deposits can reach several tens of meters, where silty clays from the Champlain Sea are covered with sandy materials (IRDA, 2008).





Regional fractured bedrock aquifers flow from the Appalachians to the St. Lawrence River (south/southeast to

north/northwest). The aquifers are moderately productive and shallow, with the water table 3 m to 5 m from the surface (Meyzonnat et al., 2018). They are in unconfined conditions upstream, where bedrock outcrops or under coarse superficial deposits, and reach semi-confined to confined conditions once the bedrock is covered by finer sediments in the valleys and close to the St. Lawrence lowlands (Carrier et al., 2013; Larocque et al., 2013, 2015b; Lefebvre et al., 2015; McCormack and Therrien, 2014; Rivard et al., 2014; Talbot-Poulin et al., 2013). Limited-extent shallow aquifers have been identified in the

local superficial coarse deposits. Previous local or watershed-scale recharge studies have identified recharge areas upstream, mostly on the outcropping bedrock and in coarse glacial sediments, while the St. Lawrence tributaries drain groundwater over most of the study area. Gagné et al. (2018) showed that in the main GWR zones, the hydrogeological system has a rapid response and most of the GWR rapidly reaches the nearby drainage system. The overall configuration corresponds to topographic-driven water tables (Gleeson et al., 2011), with hydrogeological watersheds matching superficial watersheds, and

can reasonably be considered to have a closed watershed water budget.

Daily flow rates are available from 51 gauging stations spread throughout the eight main tributary watersheds (CEHQ, 2019; Fig. 1), with watershed areas ranging from 30 to 3 000 km$^2$. Data missing for up to five consecutive days were linearly interpolated to optimize the length of the available time series. These 51 stations all have full-year measurements for more than three consecutive years, and the presence of ice most likely affects winter flow measurements.

**3 Method**

**3.1 HydroBudget**

HydroBudget (HB) is a spatially-distributed GWR model that computes a superficial water budget on grid cells of regional-scale watersheds with outputs aggregated into monthly time steps. The model uses commonly available meteorological data (daily precipitation and temperature, spatialized if possible) and spatially-distributed data (pedology, land cover, and slopes).

It is based on simplified process representations and is driven by eight parameters that need to be calibrated (Table 2). Coded in R, HB uses a conceptual lumped reservoir to compute the soil water budget on a daily time step. For each grid cell and each time step, the calculation distributes precipitation as runoff, evapotranspiration (ET), and infiltration that can reach the saturated zone if geological conditions below the soil allow deep percolation to occur (potential GWR), with a monthly time step (Fig. 3). The model was initially developed by a team at Université du Québec à Montréal to compute interannual spatially-

distributed potential GWR at the regional scale (Larocque et al., 2013, 2015a, 2015b). A full description of the model, including a comparison with other water budget models, is available in the model user guide (Dubois et al., 2021).

Calculations begin with the vertical processes, by first determining the available liquid water (vertical inflow, VI), which is the sum of rain and snowmelt, with a 0°C air temperature threshold for rain or snow precipitation and a two parameter ($T_M$ and $C_M$) degree-days approach (Massmann, 2019). If the soil is frozen (two parameters, $TT_F$ and $F_T$), the entire VI will produce

runoff directly. Otherwise, VI can runoff, infiltrate, evapotranspire, and eventually percolate as potential GWR.





Runoff is calculated using the runoff curve number (RCN) method (USDA-NRCS, 2004; 2007) on a cell-by-cell basis (two parameters, $t_{API}$ and $f_{runoff}$), similar to what is done in SWAT (Arnold et al., 2012; Neitsch et al., 2002). Vertical inflow that has not runoff is considered to be infiltration ($I$) reaching a soil reservoir for which storage capacity is driven by one parameter ($sw_m$). If $I$ exceeds available storage, saturation excess is added to runoff. Otherwise, actual evapotranspiration (AET) is

calculated as the minimum of PET (calculated using the formula of Oudin et al. (2005)) and the available water in the soil reservoir. The residual soil water is mobilized as potential GWR (one parameter, $f_{inf}$) or stored for the next time step. HydroBudget thus calculates potential GWR, the recharge that could reach the aquifer) if 1) the geological material below the soil horizon allows deep percolation to occur, 2) no additional storage or losses occur in the unsaturated zone below the soil, and 3) no significant groundwater evapotranspiration occurs (Doble and Crosbie, 2017). Actual GWR corresponds to the part

of potential GWR that would actually reach the water table, and potential GWR is therefore a maximum.

### 3.2 Calibration strategy

The HydroBudget model is calibrated on superficial watersheds, based on the hypotheses that: 1) surface watersheds match hydrogeological watersheds, 2) the rivers drain unconfined aquifers, and 3) the watershed response time is shorter than one month, thus compensating for the absence of water routing. Under these conditions, for any given watershed, monthly potential

GWR should be similar to monthly river baseflow at the outlet, and the sum of monthly runoff and monthly potential GWR should be equal to the total flow at the outlet (although monthly flows are considered, daily time steps are used in the calculations).

The daily flow rates at the outlets were those from the 51 gauging stations (Fig. 1). Baseflows were estimated from the flow rate time series following the proposition of Ladson et al. (2013) of a standard approach of the Lyne and Hollick filter (Lyne

and Hollick, 1979), using a stochastic calibration and 30 filter passes. Total flows and baseflows were divided by the area of the given watershed to provide flow values in mm/yr, and thus facilitate the comparison of calibration results between watersheds of very different sizes.

The model was simultaneously calibrated for all the gauging stations using the automatic calibration procedure of the R package *caRamel* (Monteil et al., 2020) to obtain a regionalized set of parameters. The eight HB parameters were optimized

for each gauging station, grouped by river watershed to save computational time (from 51 individual optimizations to eight grouped optimizations), and averaged over the study area using the normalized density of stations per group (number of stations per km$^2$) as weights. Up to 5 000 model calls were used per station group, with several successive optimizations to confirm the reproducibility of the results, as recommended by the algorithm's authors. Model performance was assessed using the Kling–Gupta Efficiency (KGE) calculated for monthly measured river flows and simulated total flow ($KGE_{qtot}$), as well as

monthly baseflow and monthly potential GWR ($KGE_{qbase}$). The KGE is preferred here because it better represents the low flow period than the Nash-Sutcliff Efficiency (Gupta et al., 2009). Each river flow time series was divided into a calibration period (first two thirds) and a validation period (last third), therefore allowing the objective functions to be computed per period.



The *caRamel* algorithm (Monteil et al., 2020), a combination of the multi-objective evolutionary annealing simplex algorithm (MEAS; Efstratiadis and Koutsoyiannis, 2008) and the non-dominated sorting genetic algorithm II (ε-NSGA-II; Reed and

Devireddy, 2004), automatically calibrated the eight HB parameters to maximize $KGE_{qtot}$ and $KGE_{qbase}$ values. The algorithm produces an ensemble of parameter sets (called *generation*) to run the model, downscales the generation to the parameter sets that optimize the objective functions, and creates a new set of parameters that produces better results. To produce new generations and ensure that the optimization tends toward a global maximum, the algorithm samples the parameter sets that individually maximizes the two objective functions, $KGE_{qtot}$ and $KGE_{qbase}$, samples the parameter sets that maximizes the

minimum values of the two objective functions, and increases the variance of each parameter. Finally, the best compromise for each group of gauging stations was chosen by identifying the highest mean KGE value ($KGE_{mean}$):

$$KGE_{mean} = 0.4 \times KGE_{qtot} + 0.6 \times KGE_{qbase} \qquad (1)$$

The weights attributed to each objective function in $KGE_{mean}$ were arbitrarily chosen to select the calibrated parameter set that maximizes the reproduction quality of the baseflows, considered to be the proxy for GWR ($KGE_{qbase}$), without losing the

benefits of the multi-objective optimization.

**3.3 Model sensitivity**

Using the R package *Sensitivity* (https://CRAN.R-project.org/package=sensitivity), the Morris global sensitivity approach (Morris, 1991), enhanced by Campolongo et al. (2007), was tested for the W6 (Bécancour) group of gauging stations and for the two objective functions, $KGE_{qtot}$ and $KGE_{qbase}$. The analysis consists of randomly and individually changing the eight

calibration parameters of the model (one model call per change) until all have been changed and starting again from a new combination of random values to measure the individual effects of the tested parameters, known as the elementary effects. A repetition represents one model call for an initial set of parameters followed by eight model calls, one for each parameter. A large number of repetitions (> 20) limits the convergence of the sensitivity measurement toward a local maximum. The *n* number of repetitions produces *n* number of elementary effects per parameter, the absolute values of which are averaged ($\mu*$)

to sort the parameters from most (highest $\mu*$ value) to least sensitive (lowest $\mu*$ value). The standard deviation of the elementary effects per parameter ($\sigma$) is used to estimate the linearity (limited $\sigma$ value) or non-linearity/probable interactions of the parameter with other parameters (when $\sigma$ increases; Iooss and Lemaître, 2015).

**4 Results**

**4.1 Model sensitivity and model calibration**

The per parameter sensitivity for the W6 (Bécancour) group of gauging stations was obtained with 60 repetitions of the design (540 model runs). The $\mu*$ (mean of the absolute values of the elementary effects) of both objective functions (Table 2) shows that total flow, related to $KGE_{qtot}$, is mainly sensitive to the snow melting temperature and the snowmelt coefficient ($T_M$ and



$C_M$), and to a lesser extent to the runoff correction factor ($f_{runoff}$). Potential GWR is more sensitive to parameter variations, with the runoff correction factor (influencing the partitioning of runoff and infiltration into the soil reservoir− $f_{runoff}$) being the most

sensitive parameter, followed by percolation from the reservoir into the unsaturated zone ($f_{inf}$), the soil freezing temperature ($TT_F$), the evolution from wet to dry soil conditions ($t_{API}$), and the soil water content capacity ($sw_m$). Based on their relatively high $\sigma$ values (the standard deviation of the elementary effects), these parameters might have non-linear effects and interactions on the simulated potential GWR and total flow. The time to freeze the soil ($F_T$) was found to have a negligible effect on the model outputs, and could therefore be set to a fixed value (16.4 d; the mean of the calibrated values from the first set of

optimizations before it was removed from the calibration procedure) for the final sets of calibration.

Up to 5 000 model runs were performed to calibrate the W6 (Bécancour) group of stations, and the 25 best compromises following Eq. 1 were extracted to compare the equifinality in the solutions. The KGE$_{mean}$ (before regionalization) varies within a range of ±0.005 in calibration and ±0.02 in validation (supplementary Table A. 1). Furthermore, the variability of the optimized parameters of the 25 best compromises is very limited (supplementary Table A. 1), therefore allowing to assume

that the optimization per gauging station group converged and is complete. Although HB calibration for a given group of stations was relatively straightforward because of the relatively short simulation time (e.g., 10 min for a 6 750 km² watershed, 500 m resolution grid (27 000 cells), for 58 years with 15 cores and 50 GB of RAM), based on the convergence quickly obtained for W6 (before 1 000 model runs), and the reproducibility of the results found by restarting the optimization several times, calibration of the rest of the gauging station groups were undertaken with 1 500 model runs in order to save

computational time.

Following regionalization, a satisfactory fit was observed between calibrated model and measured flow rates for all gauging station groups (Table 3). The KGE$_{mean}$ varied between 0.64 and 0.82 (average of 0.72) for the calibration period and between 0.62 and 0.77 (average of 0.69) for the validation period. The objective functions are very similar between the two periods, confirming the satisfactory calibration of the model and its capacity to reproduce the water budget over a long period,

considering that calibration and validation span 1961 to 2017, and for the entire study area.

### 4.2 Simulated potential groundwater recharge for 1961-2017

The calibrated HB model was used to simulate potential GWR for the entire study area on a 500 m x 500 m grid for the 1961-2017 period. Examples of the resulting monthly vertical inflows, river flow rates, baseflows, potential GWR, and AET are illustrated in Fig. 4 for the downgradient station in W8 for the 2000-2010 (validation) period. The simulated maximum monthly

total flow rates were close to the observed maximum values (observed values vary from 140 mm/month to 277 mm/month, simulated values vary from 139 mm/month to 356 mm/month), and these occurred simultaneously every year in the spring month(s) of maximum VI. The model generally reproduced lower river flow rates during summer, as a result of high evaporation. Observed and simulated summer/fall and winter river low flows were of similar amplitudes (in August, September, and October observed values vary from 4 mm/month to 28 mm/month and simulated values vary from

0.7 mm/month to 37 mm/month; in January and February observed values vary from 4 mm/month to 34 mm/month and





simulated values vary from 4.5 mm/month to 30 mm/month) and occurred simultaneously. In most years, river flow increased again in November and the model again reproduced this behaviour (observed values vary from 12 mm/month to 91 mm/month, simulated values vary from 26 mm/month to 85 mm/month). Simulated AET was null in the winter until the spring thaw, after which it quickly reached its highest value (98 mm/month to 122 mm/month) in July and decreased at the end of August,

reaching null values again in November. The simulated potential GWR compared favorably with baseflows estimated using the Lyne and Hollick (1979) digital filter, with maximum values reached simultaneously in April (observed values vary from 21 mm/month to 64 mm/month, simulated values vary from 12 mm/month to 45 mm/month), lowest values in August-September (observed values vary from 2 mm/month to 10 mm/month, simulated values vary from 0.2 mm/month to 6.1 mm/month) and February (observed values vary from 3 mm/month to 18 mm/month, simulated values vary from

4.4 mm/month to 18 mm/month). A second and usually less important baseflow and GWR peak was observed and simulated in November-December of most years. Similar results were obtained for the other gauging stations and for the rest of the study period for this gauging station (not presented here).

The distribution of GWR rates showed a zone of low GWR rates (< 140 mm/yr) in the western part of the study area (flat and clayey St. Lawrence Platform), mainly in W1, W2 (except for its southeast part), the central and northwestern part of W3, and

the northern part of W4, W5, W6, and W7 (Fig. 5a). A zone with higher GWR rates (140 mm/yr to 280 mm/yr, locally > 280 mm/yr) covered the southern part of W3, W4, W5, W6, and W8 (Appalachians). Areas of high GWR rates in the central part of the study area (W2, W3, W4, W5, and W6) also corresponded to the zones of higher precipitation (Fig. 2c). The regional GWR/precipitation ratio (Fig. 5b) identifies the areas of preferential recharge with ratios higher than 0.15, corresponding to areas of potential GWR rates higher than 140 mm/yr mainly located in the Appalachians (southern part of

W2 and W3, and W4 to W8). In the St. Lawrence Lowlands (W7 and lower portions of W4, W5, W6 and W8) and on the Appalachians Piedmont (central portions of W3), the ratios above 0.2 are associated with superficial coarse materials disconnected from the regional water table, meaning that the simulated potential GWR probably does not reach the fractured bedrock aquifer.

### 4.3 Partitioning and seasonality of the water budget

Across the study area, the average simulated runoff was 444 mm/year, varying between watersheds from 368 mm/yr (W1) to 485 mm/yr (W4). The average simulated AET was 501 mm/yr, ranging between 482 mm/yr (W1) and 512 mm/yr (W4), while the average potential GWR was 139 mm/year, ranging between 109 mm/yr (W1) and 154 mm/yr (W7) (Table 4, Fig. 6). On a watershed basis, simulated runoff corresponded to 41% of the total precipitation on average (39% in W1 to 42 % in W4, W6, W8), AET corresponded to 47% of precipitation on average (45% in W4, W6, and W8 to 50% in W1), and GWR recharge

accounted for 12% of precipitation on average (11% in W1 and W2 to 13% in W4, W5, W6, W7, and W8) (Fig. 6). Mean annual GWR rates accounted for less than 140 mm/yr in watersheds mainly located in the St. Lawrence Platform (W1, W2, and W3 to a certain extent) and for more than 140 mm/yr for the watersheds mainly located in the Appalachians (W4 to W8), according to the two zones of high and low GWR rates previously described.





The seasonality of the water budget shows clear differences between the eight watersheds (Table 4). Seasonal runoff was
divided into 10% during winter (west-east decrease from 13% for W1 to 6% for W7), 51% during spring (snowmelt effect,
west-east increase 47% for W1 to 56% for W7), 17% during summer, and 23% during fall (variations of 1 to 2% for both).
Winter AET accounted for 1% of annual AET (≤2% for all watersheds), the proportion of spring AET decreased from west to
east (between 30% in W1 and 24% in W8), while the proportion of summer AET increased from west to east (between 48%
in W1 and 57% in W8) and fall AET was between 18 and 20% for all watersheds. A similar pattern was observed for potential
GWR, with west-east gradients: winter GWR decreased from west to east, from 38% (W1) to 26% (W7); spring GWR
remained relatively constant (44% on average), and both summer and fall GWR increased from west to east, from 3% (W1)
to 10% (W8) and from 14% to 21% (W6 and W8) respectively. These variations in seasonal proportions of each variable were
driven by the regional temperature gradient (decreasing temperature from west to east).

Partitioning of runoff, AET, and potential GWR according to soil type showed maximum runoff rates for organic deposits,
often associated with wetlands, and minimum rates over clay (Fig. 7a). The highest AET rates occurred over coarse sediment
and the lowest rates over clay, while the highest potential GWR rates were found in coarse deposits and the lowest for clayey
areas. Steepening slopes produced more runoff and a slight increase of potential GWR (not for slopes > 8%) (Fig. 7b). The
increase in GWR rates associated with steeper slopes can be linked to the presence of coarser material usually corresponding
to glacial deposits on hillsides (high infiltration rates). Low runoff rates were associated with the flat and clayey areas of the
St. Lawrence Platform, where annual precipitation is the lowest of the study area (Fig. 2c). A clear contrast was visible in the
effect of land cover on the water budget (Fig. 7c), with the highest runoff rates over wetlands and lowest in forested areas. The
highest AET rates were found for wetlands, while the lowest were found for urban areas. The highest potential GWR rates
were associated with forested areas, and the lowest with wetlands.

**4.4 Temporal evolution of the simulated water budget since the 1960s**

HydroBudget simulated the temporal evolution of the water budget in the study area since the 1960s, thus producing an
exceptionally long simulated time series of runoff, AET, and potential GWR for the area (Fig. 8). Annual rainfall varied
between 845 mm/yr (1968) and 1 330 mm/yr (1990), while average air temperature varied between 3.9°C (1972) and 7.1°C
(2012) (Fig. 8a). The effect of interannual variability in precipitation was clear in the simulated runoff, which varied between
273 mm/yr (1980) and 630 mm/yr (2011), with minimum runoff rates of < 350 mm/yr for some of the driest years (i.e., 1961,
1964, 1965, 1967, 1968, 1970, 1980, and 1988) and > 550 mm/yr for some of the wettest years (i.e., 1973, 1976 1990, 1996,
2005, 2006, 2008, and 2011) (Fig. 8b). The simulated AET varied between 453 mm/yr (1978) and 563 mm/yr (2006), with
minimum AET rates < 465 mm/yr (1965, 1968, 1975, 1978, 1982, and 1989) and maximum AET rates > 540 mm/yr (1976,
1981, 2006, 2008, and 2013) (Fig. 8c). The average potential GWR varied between 89 mm/yr (1968) and 198 mm/yr (1983),
with low rates (< 110 mm/yr) in 1961, 1964, 1968, 1982, 1987, 2001, 2012, and 2015 and high rates (> 180 mm/yr) in 1983,
1990, 1993, and 2006. The potential GWR appears to be more highly influenced by precipitation than by temperature variations
(Fig. 8c).





Significant VI increase (p < 0.05 on Mann-Kendall test) was observed during winter for all the watersheds (from December to February), in the summer (from June to August) for almost all of them (except W1 and W7), and annually for all except W5 and W7 (Table A. 2 in Supplementary Material). Significant seasonal temperature increases were also observed for all seasons
in all watersheds, translating into a global interannual warming trend. Significant annual and seasonal precipitation trends, observed for the entire study area, produced significant positive trends in simulated runoff in the model (Table A. 2 in Supplementary Material), except in the spring and fall, when a few positive trends that do not exist in rainfall were measured for simulated runoff. Significant positive annual and seasonal temperature trends (in the winter and in the spring) correspond to the annual and seasonal trends in simulated AET, while the positive trends in summer and fall did not transmit to the AET.
The significant trends in winter precipitation and temperature led to positive trends in simulated winter potential GWR throughout the study area. It is interesting to note that although rising temperatures were observed throughout the simulation period (1961-2017), no decreasing trends were simulated for either runoff, AET, or potential GWR.

## 5 Discussion

### 5.1 Regional groundwater recharge

Although river flows are well monitored by an extensive gauging station network with long periods of measurement, groundwater level monitoring has been initiated in the Province of Quebec only at the turn of the century (MELCC, 2020). Recharge estimates from these time series are yet to be produced. Moreover, a regional-scale lysimeter network still needs to be implemented. Ground truthing of HB simulation results is thus limited by the lack of spatially-distributed GWR estimates from alternative field-based measurements, restricting the verification of model results to comparisons with local field and
modeling studies.

Potential GWR in the study area (139 mm/yr; 12% of annual precipitation) is comparable to that found in previous studies in the region (Table A. 3 in the Supplementary Material). For example, Chemingui et al. (2015) used an integrated hydrological model (CATHY; Camporese et al., 2010) in W1 to estimate GWR to be 200 mm/yr. This value is somewhat higher than that obtained using HB in the same area (109 mm/yr), but the resulting preferential recharge areas located close to the Canada-
USA border are similar with both approaches (i.e., 70 mm/yr to 250 mm/yr in Chemingui et al. (2015) and 70 mm/yr to 280 mm/yr with HB). In the same area, Levison et al. (2014) calibrated GWR at between 0 mm/yr (clay deposits) and 329 mm/yr (fractured bedrock) for use in a groundwater flow model. These values span a larger range than those found with HB (70 mm/yr to 280 mm/yr).

In the Appalachian Piedmont region of W3, Larocque and Pharand (2010) used a spatialized soil water budget (AgriFlux;
Banton et al., 1993) to estimate an average GWR of 215 mm/yr, with high spatial differences (between 0 mm/yr in clay to > 300 mm/yr in fractured bedrock). These values are higher than the average GWR simulated using HB (~150 mm/yr) for the same area, but are coherent with the highly variable simulated rates (< 140 mm/yr in the St. Lawrence Lowlands to 140 mm/yr to > 210 mm/yr in the Appalachians).





The average recharge rates simulated using the HELP model (Schroeder et al., 1994) in some watersheds of the study area are
similar to the HB-simulated potential GWR rates from this study: 86 mm/yr for W1 (Croteau et al., 2010) compared to
109 mm/yr with HB; 100 mm/yr for W2 and W3 (Carrier et al., 2013) compared to 119 mm/yr and 139 mm/yr with HB;
166 mm/yr for W8 (Lefebvre et al., 2015; and ~150 mm/yr for the lower part of W8 (Talbot-Poutin et al., 2013)) compared to
145 mm/yr with HB. Using the WaterGAP Global Hydrology Model (WGHM; Döll et al., 2003) and a 0.5° spatial resolution,
Döll and Fiedler (2008) found GWR rates ranging from 100 mm/yr to 300 mm/yr in southern Quebec, thus generally matching
360    the range of HB-simulated GWR. Wada et al. (2010) simulated much higher GWR rates over the study area (300 mm/yr to
1 000 mm/yr) with the global hydrological model PCR-GLOBWB (Bierkens and van Beek, 2009) at a 0.5° spatial resolution,
which do not match HB estimates, nor other results for the study area.

The spatially-distributed potential GWR values show a clear difference between areas in the St. Lawrence Platform
(GWR < 140 mm/yr) and areas in the Appalachian geological units (GWR > 140 mm/yr). Higher recharge occurs when soils
are coarser or in outcropping bedrock areas, which are mostly located in the Appalachians and upgradient in the watersheds,
in accordance with what has been found in previous studies (Carrier et al., 2013; Croteau et al., 2010; Larocque et al. 2013,
2015b; Lefebvre et al., 2015; Talbot-Poutin et al., 2013). Spatial variations in GWR also correlate well with land use, with
preferential infiltration zones in forested areas and lower infiltration rates over wetlands and urban areas. When associating
GWR with land cover in Belgium (oceanic climate), Zomlot et al. (2015) showed that recharge preferentially occurs in
shrublands and forests (with a classification for different forest types), while the lowest GWR is expected over urban land
cover (city center, build up, and industry) and agricultural zones (except for land cultivated for maize and tubers). Similar
results were obtained in the current study. Overall, local superficial conditions, such as soil type and land use, mostly influence
GWR rates. Similar results of factors influencing GWR are reported by Batelaan and De Smedt (2007) and Zomlot et al. (2015)
in Belgium, and by Nielsen and Westenbroek (2019) in Maine (USA).
Actual GWR is most likely lower than potential GWR, especially where superficial deposits are thick and have low hydraulic
conductivity, or where impermeable sediments are present at depth and may induce confined conditions for underlying
aquifers. To overcome this, Rivard et al. (2013) and Gagné et al. (2018) applied a 40% reduction coefficient in the semi-
confined areas of W5 to estimate actual GWR from potential GWR and assumed that no recharge occurred over confined
aquifers.

**5.2 Temporal patterns of groundwater recharge**

The annual variability in potential GWR is closely linked to that of total precipitation, with maximum variation between the
average annual rates of 100 mm/yr, between 89 mm/yr (1968) and 198 mm/yr (1983). Potential GWR occurs mainly in the
spring (44%) and winter (27% to 38%), while almost no GWR occurs in the summer (3 % to 10%), and a moderate GWR
contribution to yearly recharge in the fall (14 % to 21%). Differences between watersheds in this seasonal dynamic seem to be
driven by the southwest-northeast decrease in temperatures, with higher winter GWR rates in the warmer western watersheds
and higher summer and fall GWR in the colder eastern watersheds. Similar observations can be made for runoff and inversely





for AET. This spatiotemporal link between GWR and precipitation and temperature patterns, one of the novel contribution of this work, is coherent with that reported in other studies, both in similar and different climate and geological environments (Abdollahi et al., 2017; Ashaolu et al., 2020; Chemingui et al., 2015; Fu et al., 2019; Hayashi and Farrow, 2014; Hu et al.,

2019). GWR is inversely correlated with AET, occurring preferentially when AET is low, immediately following snowmelt and before the onset of soil frost. Other studies in the region also found spring to be the main recharge period in most years. Larocque and Pharand (2010) showed that 75% of recharge occurs in April in W3, while studies using the HELP model showed that recharge events in the spring and fall account for more than 70% of the annual recharge (Carrier et al., 2013; Croteau et al., 2010; Guay et al., 2013; Lefebvre et al., 2015; Talbot-Poutin et al., 2013). A similar high proportion of winter GWR

associated with winter precipitation and with the preferential spring recharge period have been identified by Jasechko et al. (2017) in central Canada based on isotopic analyses, and worldwide with preferential infiltration periods during the cold months (Jasechko et al., 2014).

The increasing trends in simulated winter potential GWR are most likely related to that of winter temperature and precipitation between 1961 and 2017. An original contribution of this work is to show that the absence of decreasing trends in GWR despite

the overall statistically significant increases in temperature and AET indicates that the increase in precipitation (vertical inflows in Table A. 2) is large enough to compensate for the increases in AET. As a result, GWR probably increased in southern Quebec over the 1961-2017 period. However, this cannot be verified with *in situ* data because a reliable and continuous groundwater level observation network was only implemented in southern Quebec around the turn of the century. Similar results were obtained with a groundwater flow model in the upper W1 by Levison et al. (2016), who found a non-significant

positive trend for simulated baseflows for the 1966-2010 period. However, Rivard et al. (2009) found no significant temporal trends in mean baseflows for southern Quebec for the 1956-2005 period, but rather some decreasing trends in minimum baseflows.

## 5.3 Implications for water management

This study has shown that the use of a GWR water budget model easily produced long-term spatially-distributed GWR values

at the regional scale, with an acceptable spatial resolution and monthly time steps. The average GWR rate for the region, based on the new regional estimate, is 139 mm/yr, with lower aquifer renewal rates in the St. Lawrence Platform (< 140 mm/yr) than in the Appalachians (> 140 mm/yr). Preferential infiltration areas correspond to forested area (170 mm/yr) and areas covered by coarse sediments (180 mm/yr) or outcropping bedrock (150 mm/yr). The decreasing temperature from west to east impacts the intra-annual GWR dynamic, imposing higher winter GWR in the warmer watersheds, from 38% of annual recharge in W1

to 27% in W8. Inversely, summer and fall GWR rates are lower in the western watersheds (3% and 14% respectively in W1) than in the eastern watersheds (10% and 21% respectively in W8). Furthermore, the warming temperatures since 1961 have not negatively affected GWR until 2017, due to a simultaneous increase in precipitation and higher winter GWR, compensating for the increase in AET. However, with more intense and faster climate change, this dynamic could change in the near future if temperature warming was to exceed precipitation increase.





Based on the study results, GWR maps should be included in land management planning to avoid impacting areas of preferential recharge, such as forests, areas covered by coarse material, and zones of outcropping bedrock. Although the GWR/precipitation ratio in eastern North America seems to be 0.25 and higher (Jasechko et al, 2014, based on the global simulation of Wada et al. (2010)), areas of preferential recharge can be identified in the study area based on a GWR/precipitation ratio higher than 0.15 (the worldwide mean in Jasechko et al., 2014). These represent essential areas for

the quantitative replenishment of groundwater resources, and therefore the most vulnerable areas in terms of water quality of southern Quebec. Forested areas should be prioritized for future preservation, and forest conservation policies implemented for groundwater resource protection. Although it has been shown that wetlands are often connected to the water table in southern Quebec (Bourgault et al., 2014) they are not considered to be preferential infiltration areas in HB. Separate studies will be necessary to investigate their role in regional groundwater flow. Considering the substantial influence of winter on the

regional hydrology, one of the novel outcome of this study, and that changes in the winter dynamic could highly influence the entire hydrologic dynamic of cold regions (Jasechko et al., 2014, 2017), additional winter-related research should target gauging of winter river flows, soil frost modelling, enhancing the vertical inflow and snowpack modeling, and would probably require the development of the winter monitoring network.

**5.4 Contributions to and limitations of water budget models in groundwater recharge simulation**

HydroBudget was specifically developed to simulate spatial and temporal variations in potential GWR at the regional scale, for long periods, and for regions where winter largely affects the intra-annual hydrologic dynamic. The model can reasonably be used where hydrogeological and superficial watersheds are similar, where regional groundwater flow converges to rivers, and when a monthly resolution is sufficient. Although the model does not compute water routing, groundwater-surface water feedback, or evapotranspiration from groundwater, and produces potential GWR, the good simulation results found in this

work justify the water budget calculation scheme used in HB, resulting in an easy-to-use and computationally efficient model. Similar to WetSpass, which has been applied in different climate and geological environments (Abdollahi et al., 2017 – West Africa, semiarid to sub-humid climates; Zomlot et al., 2015 – Belgium, oceanic climate), or HELP (Croteau et al., 2010 - Quebec, continental climate Toews and Allen, 2009 – British Columbia, arid climate), HB could probably be adapted for use in different environments than that presented here.

The impact of long and cold winters was included in HB through the widely used degree-days method that represents snowpack evolution (Massmann, 2019) and the representation of freezing soil conditions with a threshold temperature and a duration of the threshold temperature to freeze the soil ($TT_F$ and $F_T$ respectively). The sensitivity analysis shows that $TT_F$ is one of the key parameters in potential GWR simulation, while $F_T$ was found not to be sensitive in model calibration (Table 2). This result underlines the importance of including soil freezing in GWR modeling for cold regions, and more specific observation data

could further enhance it.

Among the simplifications included in several water budget models (HELP; HydroBudget; SWB; water balance GIS tool; Huet et al., 2016) is the calculation of runoff using the RCN curve number method (USDA-NRCS, 2004). This method has



been criticized for its empirical basis, developed specifically for the US context (Ogden et al., 2017). However, it is continuously adapted and used for new environments and different size areas and is well-documented and easy to use (Bartlett et al., 2016; Lal et al., 2019; Miliani et al., 2011; Ross et al., 2018). Notably, it is used to estimate runoff in the SWAT hydrological model (Neitsch et al., 2002), and is calibrated specifically in the Quebec climate and geological context (Gagné et al., 2013; Monfet, 1979). In water budget models, the RCN method drives the partitioning of the simulated water budget into runoff and water available for AET and GWR, therefore making the related parameters ($t_{API}$ and $f_{runoff}$) particularly sensitive in their simulation (e.g., $f_{runoff}$ is the most sensitive parameter for potential GWR in HB; Table 2). Because it is based on land cover, topography, soil types, and seasonal humidity, the RCN method reflects the spatial variability of the large study area, which probably contributes to the ability of HB to simulate the surficial water budget and discriminate its partitioning depending on the superficial conditions (superficial geology, land cover).

The AET corresponded to 47% of the annual precipitation (501 mm/yr), similar to previous studies in the study area (Croteau et al., 2010; Carrier et al., 2013; Benoit et al., 2014; Larocque et al., 2015b; Meyzonnat, 2012; Table A. 3). Considering the vegetation cover of southern Quebec (45% of the area of forest and 6% of wetlands), the water availability, with an average of 1,090 mm/yr of precipitation evenly distributed throughout the year, the topography-driven water table, and the shallow depth of the aquifers, groundwater-vegetation-atmosphere exchanges are probably intensive (Koirala et al., 2017; Xu and Liu, 2017). The model configuration, with a soil lumped reservoir for AET-potential GWR partitioning, is therefore conceptually suitable (Cuthbert et al., 2019). AET computation is driven by two parameters, i.e. the runoff factor ($f_{runoff}$), influencing the partitioning between runoff and infiltration into the soil reservoir, and the soil reservoir parameter ($s_{wm}$), determining the reservoir capacity. Evapotranspiration calibration data would be useful to better constrain these two parameters.

The simulated transient GWR was calibrated with baseflows computed using regressive filters from river flow rate time series. Although Partington et al. (2012) showed that the association between baseflows and GWR is not always satisfactory, baseflows are generally considered to be an acceptable proxy for GWR in Canada (Chemingui et al., 2015; Rivard et al., 2009) and widely used for the calibration of GWR simulation (Batelaan and De Smedt, 2007; Croteau et al., 2010; Dripps and Bradbury, 2007; Gagné et al., 2018; Rivard et al., 2013). Large differences in volumes and timing have been shown using different baseflow separation methods (Gonzales et al., 2009; Zhang et al., 2017). The choice of the baseflow separation method will therefore influence the proportion of baseflow in the river flow and impact the parameter calibration, the simulated potential GWR, and consequently the water budget partitioning. However, the structure of water budget models makes them compute GWR as the residual of the water budget, thus propagating the computational error from the other terms of the water budget, namely runoff, interception, evapotranspiration, subsurface runoff (depending on the model complexity) onto GWR estimation (Crosbie et al., 2015; Scanlon et al., 2002). For this reason, using baseflows as explicit calibration data for GWR simulation limits the modeling error of GWR rates, while making it dependant on the baseflow separation method.

Having a model that is fully coded in an open-source language leaves the option of reshaping its structure and conceptual model to specific study needs and environments. Changes to the model could include, for example, changing the PET calculation to a formula considered to be more suitable for a specific local context, defining a different runoff computation





method that would better represent local conditions, modifying the representation of the soil reservoir by spatializing it, or using other calibration data, such as observation of AET or other baseflow filters. However, it should be kept in mind that increasing the complexity of the process representation would increase the number of parameters to be calibrated and most

likely increase computation time as well. An original contribution of this word was to show that the tested calibration method combined with a regionalized parameter set offers a very acceptable solution that is highly reproducible and could be applied in less monitored regions. Apart from being relatively simple to use, the HB model has high potential for non-hydrogeologist users, especially if the computational time is limited and the update of observation data and the automatic calibration procedure are easy tasks. Hydrogeologist modellers might be interested in including HB as a complementary tool for comparative

purposes when using integrated models or when in need of GWR estimates for groundwater flow models.

## 6 Conclusion

Groundwater recharge is recognized as a strategic hydrologic variable that needs to be estimated for sustainable groundwater management, especially within a global change context, but is challenging to simulate at the regional scale and for long-term conditions because of long computational times and the large number of calibration data required to produce reliable results.

Such knowledge is key to the implementation of long-term water management and land use policies. The objective of this work was to test the ability of the HydroBudget water budget model to simulate long-term regional-scale GWR. The model is used in southern Quebec, where the hydrological dynamic is highly linked to harsh winters. With the model simultaneously calibrated on 51 gauging stations (river flows and baseflow estimates), GWR was reliably simulated between 1961 and 2017 at the regional scale (36 000 km$^2$). GWR estimates at this scale, including eight tributary watersheds of the St. Lawrence River

with a monthly time step and 500 m x 500 m resolution was not available until now. Water budget partitioning has proven to be highly useful to provide this information with spatiotemporal patterns and trends for regional GWR in southern Quebec. The influence of winter on the regional hydrologic dynamic, and especially the ability of the HB model to simulate snowpack evolution and soil frost appears to be a key feature of this promising approach. Nevertheless, additional research focusing on snow melting and freezing soil processes would help to better constrain the model for the winter and spring periods,

consequently helping better anticipate future changes.

A particular and especially useful outcome of this work is the water budget partitioning into runoff (41%, 444 mm/yr), AET (47%, 501 mm/yr), and potential GWR (12%, 139 mm/yr). This study shows that groundwater recharge peaks in the spring (44% of annual recharge) and is high during winter (32% of annual recharge), while the seasonal GWR distribution varies markedly between watersheds and is linked to annual temperatures. Interestingly, the long simulation period made it possible

to identify a statistically increasing trend of GWR during winter and no decreasing trends in the summer, despite warmer temperatures, most likely because the resulting increase in AET is compensated by a significant increase in precipitation. However, considering the intense and fast climate change expected in future decades, the water budget partitioning and the regional GWR could decrease if temperature warming were to exceed the increase in precipitation.





Besides being very useful for acquiring knowledge on regional GWR, a regional model that can easily be updated is extremely
useful for water management to implement water resource conservation policies, identify additional research required, and anticipate the impacts of climate change. Water budget models that are easily regionally calibrated with river flow measurements and baseflows have a high potential for non-hydrogeologist users, especially if they can easily be adapted to specific study needs and environments.

## 7 Appendices

**Table A. 1: Best compromises and optimized parameters obtained using the best compromise for the gauging stations of W6 and ranges obtained with the 25 best compromises (before regionalization).**

| | Calibration | | | Validation | | |
|---|---|---|---|---|---|---|
| | $KGE_{mean}$ | $KGE_{qtot}$ | $KGE_{qbase}$ | $KGE_{mean}$ | $KGE_{qtot}$ | $KGE_{qbase}$ |
| Best compromise | $0.76_1$ | $0.80_7$ | $0.72_9$ | $0.70_9$ | $0.71_9$ | $0.70_2$ |
| 25 best compromises | $[0.75_8 ; 0.76_1]$ | $[0.78_1 ; 0.82_{0+}]$ | $[0.71_8 ; 0.74_5]$ | $[0.69_7 ; 0.71_6]$ | $[0.70_6 ; 0.73_9]$ | $[0.67_3 ; 0.72_1]$ |

| | Optimized parameters* | | | | | | |
|---|---|---|---|---|---|---|---|
| | $T_M$ (°C) | $C_M$ (mm/°/d) | $TT_F$ (°C) | $t_{API}$ (d) | $f_{runoff}$ (-) | $sw_m$ (mm) | $f_{inf}$ (-) |
| Best compromise | 0.5 | 4.0 | -17.9 | 3.8 | 0.54 | 308 | 0.05 |
| 25 best compromises | [0.2 ; 0.8] | [3.5 ; 4.4] | [-20 ; -14.4] | [3.0 ; 4.0] | [0.52 ; 0.56] | [227 ; 439] | [0.04 ; 0.06] |

*$F_T$ is held constant at the nominal value of 16.4 d

**Table A. 2: Annual and seasonal trends for vertical inflows (rain plus estimated snowmelt), observed temperatures, simulated runoff, simulated actual evapotranspiration (AET), and simulated potential groundwater recharge (GWR) for the 1961-2017 period for the eight watersheds (Man-Kendall test; only p-values > 0.05 are presented, otherwise a "-" is used; all trends are positive).**

| | Vertical inflow | | | | | Temperature | | | | | Simulated runoff | | | | | Simulated AET | | | | | Simulated pot. GWR | | | | |
|---|---|---|---|---|---|---|---|---|---|---|---|---|---|---|---|---|---|---|---|---|---|---|---|---|---|
| | Year | Win. | Spr. | Sum. | Fall | Year | Win. | Spr. | Sum. | Fall | Year | Win. | Spr. | Sum. | Fall | Year | Win. | Spr. | Sum. | Fall | Year | Win. | Spr. | Sum. | Fall |
| W1* | 0.041 | 0.006 | - | - | - | 0.000 | 0.001 | 0.029 | 0.002 | 0.001 | 0.031 | 0.023 | - | - | - | 0.020 | 0.000 | 0.001 | - | - | - | - | - | - | - |
| W2* | 0.003 | 0.001 | - | 0.020 | - | 0.000 | 0.001 | 0.019 | 0.001 | 0.002 | 0.012 | 0.001 | - | 0.009 | 0.017 | 0.006 | 0.001 | 0.000 | - | - | - | 0.050 | - | - | - |
| W3 | 0.004 | 0.001 | - | 0.021 | - | 0.000 | 0.003 | 0.038 | 0.000 | 0.001 | 0.007 | 0.003 | - | 0.009 | 0.031 | 0.002 | 0.001 | 0.000 | - | - | - | 0.009 | - | - | - |
| W4* | 0.004 | 0.000 | - | 0.036 | - | 0.000 | 0.002 | 0.038 | 0.001 | 0.002 | 0.009 | 0.000 | - | 0.008 | - | 0.000 | 0.001 | 0.001 | - | - | - | 0.001 | - | - | - |
| W5 | - | 0.000 | - | 0.006 | - | 0.000 | 0.003 | - | 0.002 | 0.002 | - | 0.003 | 0.013 | 0.001 | - | 0.012 | 0.002 | 0.002 | - | - | - | 0.018 | - | - | - |
| W6 | 0.001 | 0.000 | - | 0.031 | - | 0.000 | 0.002 | - | 0.001 | 0.001 | 0.003 | 0.001 | - | 0.003 | - | 0.001 | 0.000 | 0.001 | 0.042 | - | - | 0.002 | - | - | - |
| W7 | - | 0.001 | - | - | - | 0.000 | 0.001 | 0.024 | 0.000 | 0.000 | - | 0.002 | - | 0.045 | - | 0.001 | 0.000 | 0.000 | - | - | - | 0.012 | - | - | - |
| W8 | 0.000 | 0.001 | - | 0.016 | - | 0.000 | 0.001 | 0.024 | 0.000 | 0.000 | 0.000 | 0.001 | - | 0.001 | 0.014 | 0.000 | 0.000 | 0.001 | 0.028 | - | - | 0.000 | - | - | - |

*Part of the watershed is located in the USA - the presented values are only for the Quebec part


**Table A. 3: Summary of GWR modeling studies considered in this work.**

| Geographic area | Reference | Runoff (mm/yr) | AET (mm/yr) | GWR (mm/yr) | Model | Seasonnality (% of annual GWR) | Temporal evolution | Influential factors |
|---|---|---|---|---|---|---|---|---|
| | | | | | **In the study area** | | | |
| W1. Châteaugay* | Chemingui et al. (2015) | - | - | 200 | CATHY (Camporese et al., 2010) | Spring = 35 % Fall = 15 % | - | - |
| | Croteau et al. (2010) | 371 | 487 | 86 | HELP (Schroeder et al., 1994) | Spring and fall = GWR peak | no significant trend found for 1963-2001 | Soil texture, low correlation GWR/precipitation |
| | Guay et al. (2013) | 267 (-) | 556 | 214 (233) | HELP (CATHY) | - | - | - |
| | Levison et al. (2016) | - | - | 113 | MODFLOW | - | Significant positive trend for ET for 1900 - 2010 Significant negative trend on GWR and baseflow for 1900 - 2010 non signigicant trend on GWR for 1966 - 2010 | - |
| | Levison et al. (2014) | - | - | 136 | MODFLOW | Spring = 64 % Fall =34 % | - | Soil texture |
| | Nastev et al. (2008) | 224 | 546 | 186 | HELP (Schroeder et al., 1994) | - | - | Soil texture |
| W3. Yamaska | Carrier et al. (2013) | 476 | 539 | 98 | HELP (Schroeder et al., 1994) | - | - | - |
| | Larocque and Pharand (2010) | 301 | - | 215 | AgriFlux (Banton et al., 1993) | Spring = 75 % Fall = 18 % | - | - |
| W5. Nicolet | Larocque et al. (2015b) | 444 | 467 | 153 | HydroBudget | Winter = 25 %, spring = 40%, fall = 33 % | - | GWR slightly impacted by precipitation variations - AET relatively constant |
| W6. Bécancour | Larocque et al. (2013) | 600 | 291 | 159 | HydroBudget | Winter = 12 %, spring = 26 %, fall = 26 % | GWR relatively constant through the years (1990-2010) | GWR slightly impacted by precipitation variations - AET relatively constant |
| | Meyzonnat (2012) | 505 | 488 | 139 | MOHYSE (Fortin and Turcotte, 2007) | Winter = 15 %, spring = 38%, fall = 30 % | - | GWR slightly impacted by precipitation variations - AET relatively constant |
| W8. Chaudière | Benoit et al. (2014) | 478 | 447 | 185 | HELP (Schroeder et al., 1994) | - | - | - |
| | Lefebvre et al. (2015) | 450 | 543 | 166 | HELP (Schroeder et al., 1994) | - | - | - |
| | | | | | **Bordering the study area** | | | |
| Vaudreuil Soulanges | Larocque et al. (2015a) | 540 | 381 | 48 | HydroBudget | Spring = 38 % Fall = 44 % | - | GWR slightly impacted by precipitation variations and not by temperature variation - AET relatively constant |
| Québec city | Talbot Poulin et al. (2013) | 350 | 550 | 445 | HELP (Schroeder et al., 1994) | Spring = 39 % Fall = 20 % | - | Soil texture |
| U.S.A. (Maine) | Nielsen and Westenbroek (2019) | - | - | 178 | SWB (Westenbroek et al., 2010) | - | - | Precipitation, soil texture, and land cover |
| | | | | | **Elsewhere in Canada** | | | |
| Ontario (Grand River) | Jyrkama and Sykes (2007) | - | - | 200 | HELP (Schroeder et al., 1994) | - | - | - |
| New Brunswick (Otter Brook) | Kurylyk and MacQuarrie (2013) | - | - | 550 | HELP (Schroeder et al., 1994) | Spring = 60 % Fall = 22 % | - | - |
| Nova Scotia (Annapolis) | Rivard et al. (2013) | 373 | 519 | 165 (115) | HELP (FEEFLOW) | - | - | Soil texture |
| British Columbia (Grand Forks) | Allen et al. (2004) | 58 | 283 | 135 | HELP (steady state) | - | - | - |
| | | | | | **Outside of Canada** | | | |
| Belgium | Batelaan and De Smedt (2007) | 49 | 465 | 251 | WetSpass | Winter > 90 % | - | Soil texture and land cover |
| | Zomlot et al. (2015) | - | - | 235 | WetSpass (Batelaan and De Smedt, 2007) | Winter = 92 % Summer = 8 % | - | Precipitation, soil texture, and vegetation cover |
| West Africa | Abdollahi et al. (2017) | 20.3%* | 65.7%* | 13.9%* | WetSpass (Batelaan and De Smedt, 2007) | All the water budget = 100 % rain season | - | rainfall, leaf area index, and PET |
| China (Chinese Loess Plateau) | Hu et al. (2019) | - | - | 18 | HYDRUS-1D (Simunek et al., 2009) | - | Significant negative trend on GWR (1981-2010) | Soil water retention, spatio-temporal precipitation patterns, soil properties, and PET and leaf area index to a certain extent |
| Worldwide | Döll and Fidler (2008) | 500 - 1,000** | - | 100 - 300** | WGHM (Döll et al., 2003) | - | - | - |
| | Wada et al. (2010) | - | - | 300 - 1,000* | PCR-GLOBWB (Bierkens and van Beek, | - | - | - |

*In percentage of the yearly precipitation
** Values extracted for the study area only

## 8 Code availability

The code of the HydroBudget model is open source and can be obtained by direct request to the authors




## 9 Author contribution

ED, ML, and SG contributed to developing the approach and all authors contributed to writing the paper. ED and SG developed the HB script in R, based on the model created by GM, which was implemented in java script. Data management, simulations, and figure preparation were done by ED. ML obtained the research grant and supervised the research.

## 10 Competing interests

The authors declare that they have no conflict of interest.

## 9 Acknowledgements

This project was funded by the Quebec Ministry of Environment and Climate Change (*Ministère de l'Environnement et de la Lutte contre les changements climatiques - MELCC*), who also provided the interpolated meteorological data.

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



## 11 Figures and tables

**Fig. 1: Location of the study area and watersheds.**





**Fig. 2: (a) Land cover (reclassified from Bissonnette et al., 2016), (b) soil types (IRDA, 2008), (c) mean precipitation (1961-2017), as well as the areas of the St. Lawrence Platform (hatched) and the Appalachians, and (d) mean temperature (1961-2017) (Bergeron, 2016).**





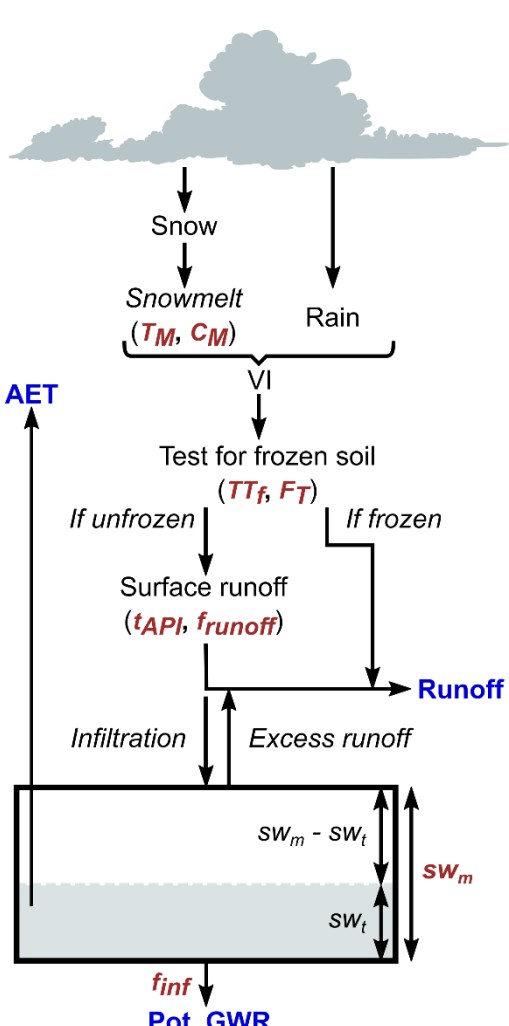


**Fig. 3: HydroBudget processes, including the eight calibrated parameters in red.**





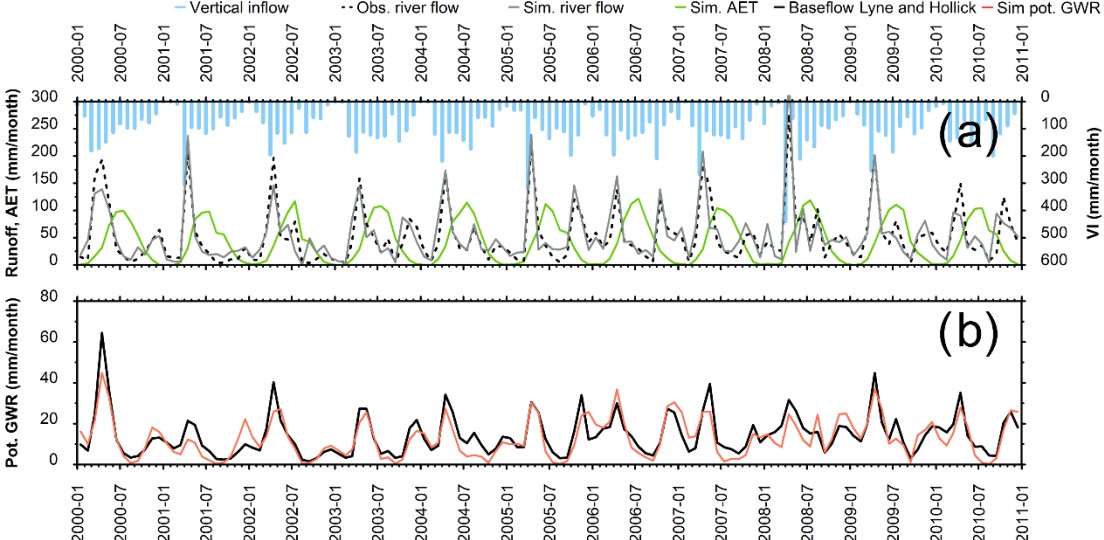

**Fig. 4: Example of a simulated water budget for the 2000-2010 period (validation period) for the downgradient station on the Chaudière River (W8) with (a) vertical inflow (VI), observed and simulated river flow rates, and simulated actual evapotranspiration (AET), and (b) baseflow (Lyne and Hollick, 1979) and simulated potential GWR.**




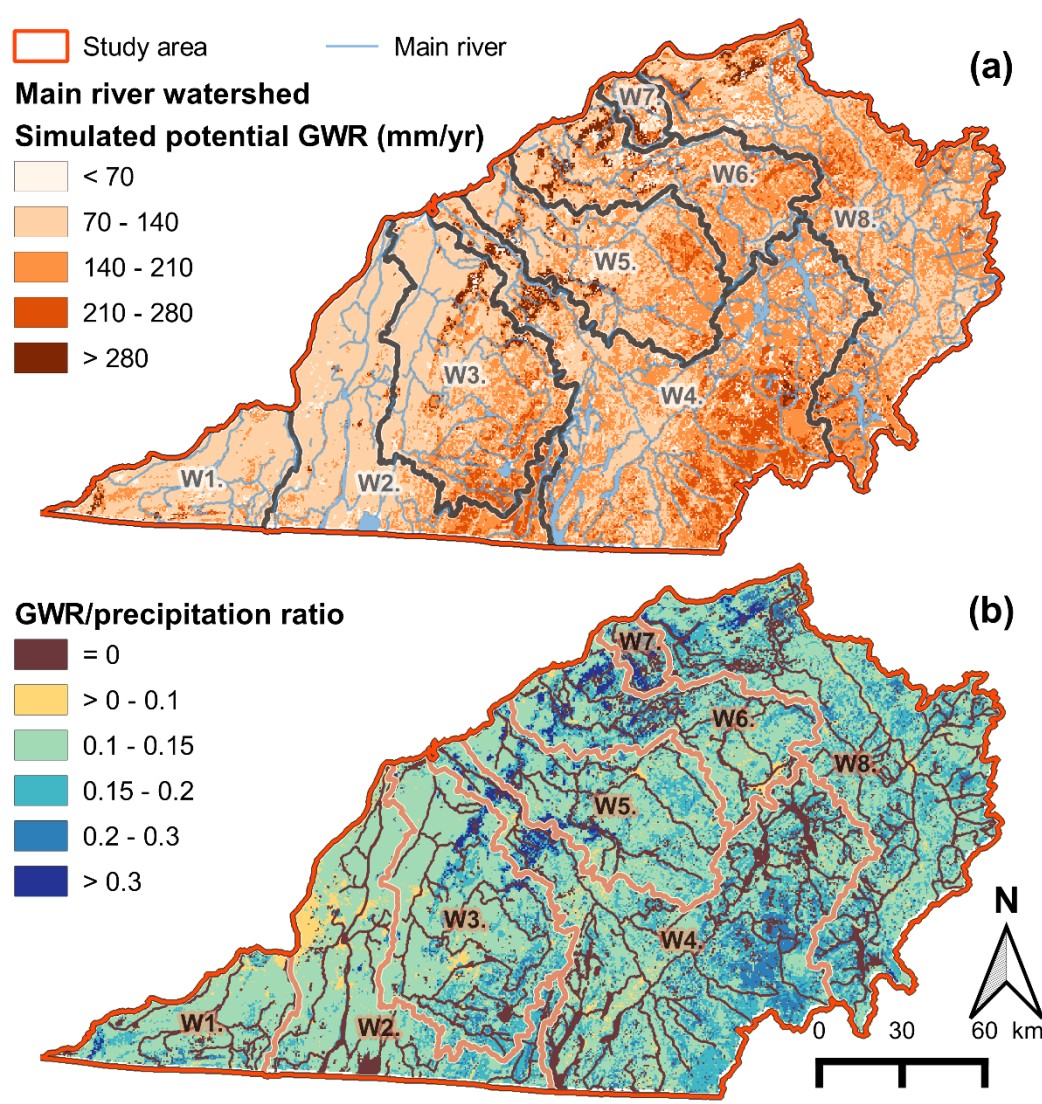

**Fig. 5: (a) Average potential groundwater recharge (GWR) simulated using the HydroBudget model for the study area between 1961 and 2017 and (b) GWR/precipitation ratio for the same period.**





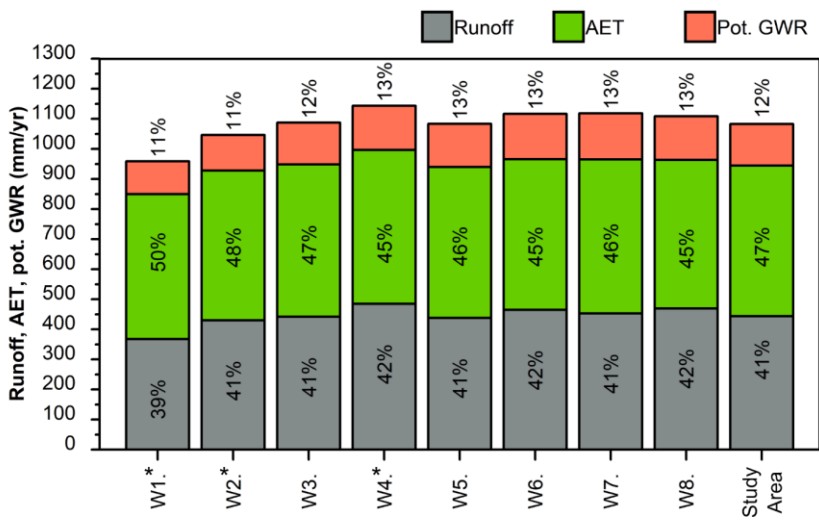

**Fig. 6: Average proportions of annual runoff, actual evapotranspiration (AET), and potential groundwater recharge (GWR) for the eight watersheds (W1 to W8) and mean water budget for the study area between 1961 and 2017. Watersheds with * are partially located in the USA (values are for the Quebec part only).**

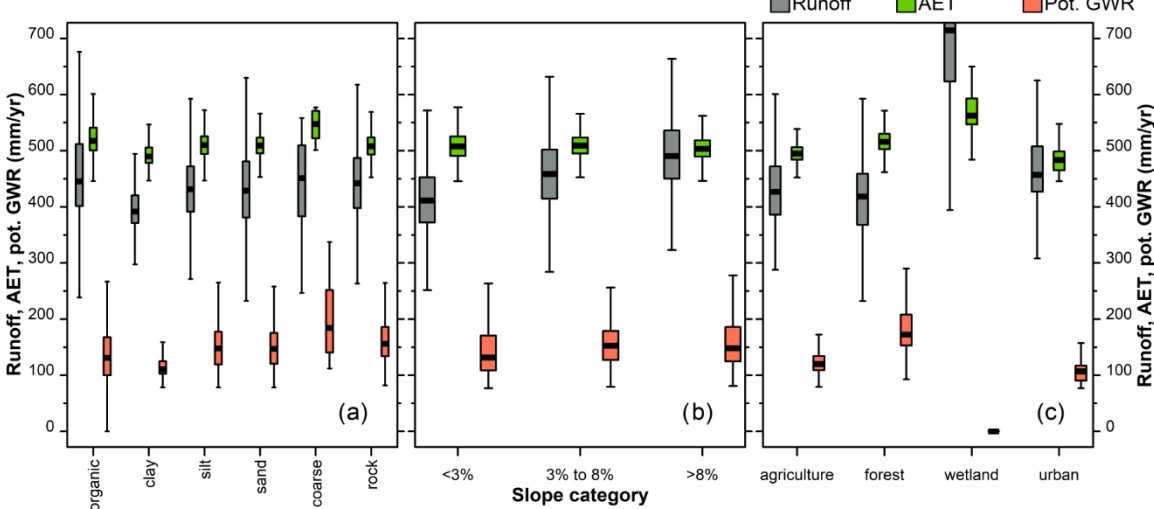

**Fig. 7: Median, 25th and 75th percentiles, minimum, and maximum values for annual runoff, actual evapotranspiration (AET), and potential groundwater recharge (GWR) throughout the study area between 1961 and 2017, classified as a function of (a) soil type, (b) slope, and (c) land cover.**



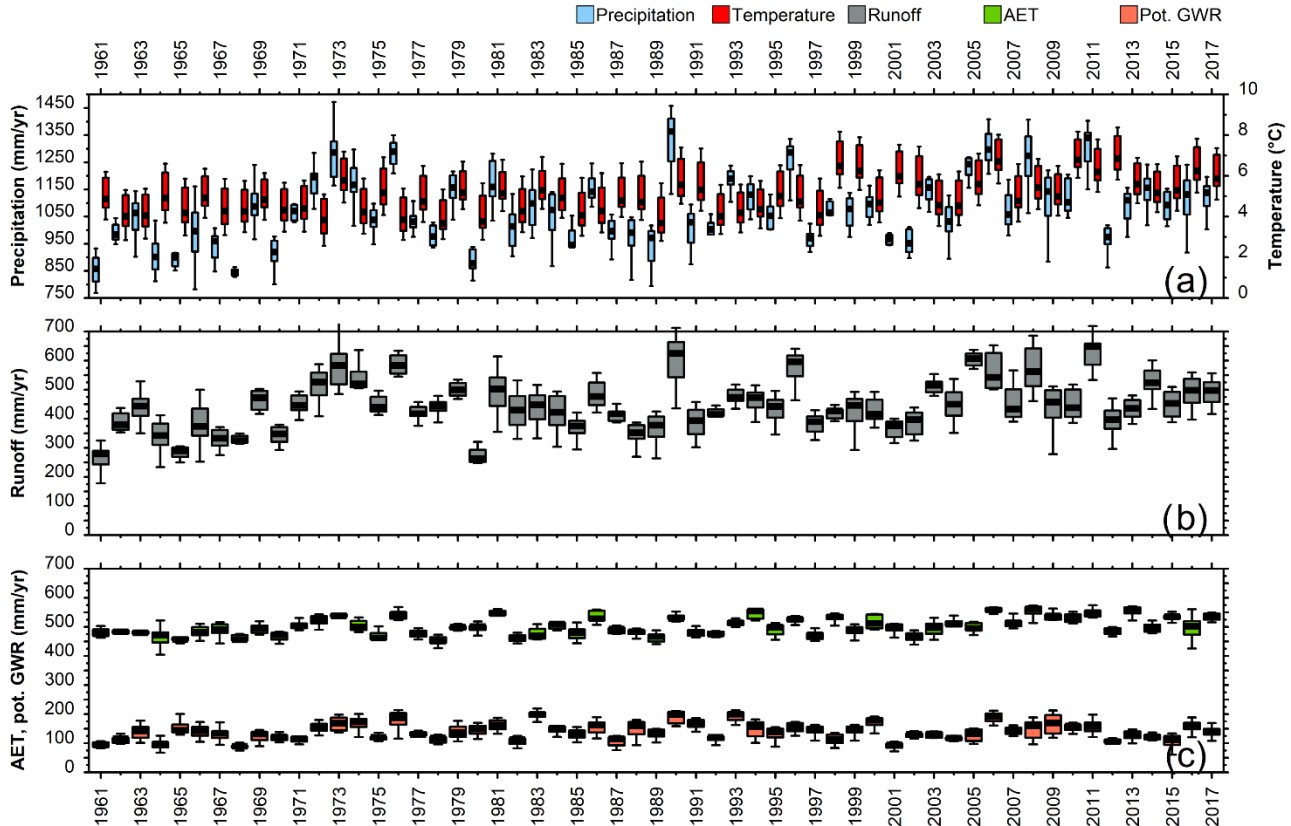

**Fig. 8: Median, 25th and 75th percentiles, minimum, and maximum fluxes of (a) temperature and precipitation, (b) simulated runoff, and (c) simulated actual evapotranspiration (AET) and potential groundwater recharge (GWR) between 1961 and 2017 for the eight watersheds.**


**Table 1: Characteristics of the studied watersheds.**

|  | Elevation (m asl) | | | Area (km²) | Land cover (% of total area) | | | | | Temp. (°C) | Precip. (mm/yr) |
|---|---|---|---|---|---|---|---|---|---|---|---|
|  | Median | Min | Max | | Agriculture | Forest | Wetland | Urban | Water | | |
| W1. Châteaugay* | 51 | 0 | 330 | 2 219 | 62 | 26 | 5 | 6 | 1 | 6.5 | 952 |
| W2. Richelieu* | 40 | 7 | 865 | 4 414 | 55 | 28 | 3 | 11 | 3 | 6.3 | 1 039 |
| W3. Yamaska | 80 | 6 | 757 | 4 792 | 57 | 34 | 3 | 5 | 1 | 5.9 | 1 080 |
| W4. Saint-François* | 312 | 4 | 1 044 | 9 068 | 23 | 62 | 7 | 4 | 4 | 4.8 | 1 123 |
| W5. Nicolet | 150 | 2 | 585 | 3 591 | 48 | 44 | 4 | 3 | 1 | 5.1 | 1 076 |
| W6. Bécancour | 140 | 4 | 652 | 3 380 | 36 | 53 | 7 | 3 | 1 | 4.5 | 1 103 |
| W7. Du Chêne | 90 | 15 | 130 | 461 | 36 | 45 | 14 | 3 | 2 | 4.5 | 1 092 |
| W8. Chaudière | 340 | 10 | 1 108 | 7 879 | 23 | 66 | 7 | 3 | 1 | 3.9 | 1 092 |

*Part of the watershed is located in the USA - the presented values are for the Quebec part only*



**Table 2: Description of the eight HydroBudget parameters and sensitivity analysis results for W6.**

| | | | | Sensitivity (W6) | | | |
| | | | | KGE$_{qtot}$ | | KGE$_{qbase}$ | |
| | Parameter | | Value range from literature | $\mu^*$ | $\sigma$ | $\mu^*$ | $\sigma$ |
|---|---|---|---|---|---|---|---|
| **Degree-days snowmelt model** | Melting temperature - $T_M$ (°C) | Air temperature treshold for snowmelt | [-2 ; 2] (Massmann, 2019) | 0.21 | 0.16 | 0.08 | 0.10 |
| | Melting coefficient - $C_M$ (mm/°C/d) | Melting rate of the snowpack | [2 ; 12] (Massmann, 2019) | 0.17 | 0.11 | 0.13 | 0.20 |
| **Freezing soil conditions** | Threshold temperature for soil frost - $TT_F$ (°C) | Air temperature treshold for soil frost | [-20 ; 0] (Henry, 2007) | 0.03 | 0.03 | 0.33 | 0.31 |
| | Freezing time - $F_T$ (d) | Duration of air temperature treshold to freeze the soil | [5 ; 30] (Henry, 2007) | 0.01 | 0.02 | 0.06 | 0.09 |
| **Runoff** | Antecedant precipitation index time - $t_{API}$ (d) | Time constant to consider the soil in dry or wet conditions based on previous precipitation event | [1 ; 5] (Lal et al., 2015) | 0.04 | 0.04 | 0.29 | 0.34 |
| | Runoff factor - $f_{runoff}$ (-) | Partitioning between runoff computed with the RCN method and infiltration into the soil reservoir | →1 (Neitsch et al., 2002) | 0.12 | 0.09 | 0.64 | 0.61 |
| **Lumped soil reservoir** | Maximum soil water content - $sw_m$ (mm) | Soil reservoir storage capacity, maximum height of water stored in a 1 m soil profile | [50 ; 900] (Croteau et al., 2010) | 0.03 | 0.04 | 0.29 | 0.37 |
| | Infiltration factor - $f_{inf}$ (-) | Fraction of soil water that produces deep percolation at each daily time step | [< 0.1 ; 1] (Croteau et al., 2010) | 0.04 | 0.07 | 0.34 | 0.48 |

**Table 3: Objective functions for the simulated outputs, for calibration and validation periods.**

| Gauging stations | | Calibration | | | Validation | | |
| Number | Measur. period | KGE$_{qtot}$ | KGE$_{qbase}$ | KGE$_{mean}$ | KGE$_{qtot}$ | KGE$_{qbase}$ | KGE$_{mean}$ |
|---|---|---|---|---|---|---|---|
| **W1*** | 2 | 1980-2013 | 0.80 | 0.65 | 0.71 | 0.79 | 0.63 | 0.70 |
| **W2*** | 5 | 1973-2017 | 0.76 | 0.56 | 0.64 | 0.75 | 0.53 | 0.62 |
| **W3** | 14 | 1965-2017 | 0.76 | 0.64 | 0.69 | 0.79 | 0.57 | 0.66 |
| **W4*** | 8 | 1961-2009 | 0.81 | 0.71 | 0.75 | 0.85 | 0.71 | 0.77 |
| **W5** | 4 | 1961-2017 | 0.77 | 0.75 | 0.76 | 0.76 | 0.59 | 0.66 |
| **W6** | 8 | 1961-2017 | 0.84 | 0.61 | 0.70 | 0.71 | 0.63 | 0.66 |
| **W7** | 2 | 1993-2017 | 0.92 | 0.75 | 0.82 | 0.86 | 0.72 | 0.77 |
| **W8** | 8 | 1961-2015 | 0.80 | 0.67 | 0.72 | 0.77 | 0.64 | 0.69 |

*the presented values are for the stations of which the watershed is completely located in Quebec*



**Table 4: Simulated runoff, actual evapotranspiration (AET), and potential groundwater recharge (GWR) for the study area between 1961 and 2017, in mm/year and in percentage for winter, spring, summer and fall, and for the eight watersheds (W1 to W8).**

| | Runoff | | | | | AET | | | | | Potential GWR | | | | |
|---|---|---|---|---|---|---|---|---|---|---|---|---|---|---|---|
| | mm/yr | Win. | Spr. | Sum. | Fall | mm/yr | Win. | Spr. | Sum. | Fall | mm/yr | Win. | Spr. | Sum. | Fall |
| **W1*** | 368 | 13% | 47% | 16% | 24% | 482 | 2% | 30% | 48% | 20% | 109 | 38% | 46% | 3% | 14% |
| **W2*** | 430 | 12% | 48% | 16% | 24% | 498 | 1% | 29% | 50% | 19% | 119 | 36% | 45% | 4% | 15% |
| **W3** | 442 | 12% | 48% | 17% | 24% | 507 | 1% | 28% | 52% | 19% | 139 | 35% | 44% | 4% | 17% |
| **W4*** | 485 | 11% | 50% | 17% | 22% | 512 | 1% | 25% | 55% | 18% | 147 | 31% | 42% | 8% | 19% |
| **W5** | 438 | 10% | 50% | 17% | 23% | 502 | 1% | 26% | 54% | 19% | 144 | 32% | 43% | 6% | 19% |
| **W6** | 465 | 8% | 53% | 16% | 23% | 501 | 1% | 25% | 56% | 18% | 151 | 28% | 44% | 7% | 21% |
| **W7** | 453 | 6% | 56% | 15% | 23% | 512 | 1% | 25% | 56% | 18% | 154 | 26% | 46% | 8% | 20% |
| **W8** | 470 | 7% | 53% | 18% | 23% | 494 | 1% | 24% | 57% | 18% | 145 | 27% | 42% | 10% | 21% |

*Part of the watershed is located in the USA - the presented values are for the Quebec part only*
*Winter: December, January, February; spring: March, April, May; summer: June, July, August; fall: September, October, November*