# Peer review of "Simulation of long-term spatiotemporal variations in regional-scale groundwater recharge: Contributions of a water budget approach in cold and humid climates"

_Hydrology and Earth System Sciences, 2021_

## Author Comment (AC1)

Fig. 1 will be modified as follows:

[Figure]

Fig. 4 will be modified as follows:

[Figure]

---

## Author Comment (AC2)

[Figure]

*Figure 1: Relative sensitivity of the eight model parameters on the simulated river flow (a) and on the simulated potential groundwater recharge (b) for the eight groups of gauging stations. The relative parameter ranking is reported in the bars from the most sensitive, 1, to the least, 8.*

---

## Author Comment (AC3)

**Table A. 1: Best compromises, interannual potential GWR, and optimized parameters obtained with the best compromises for the gauging stations of W6 and ranges obtained with the 25 best compromises (before regularization) using different calibration weights and different baseflow separation methods.**

| | Calibation | | | Validation | | | Interannual pot. GWR (mm/yr) |
|---|---|---|---|---|---|---|---|
| | $KGE_{mean}$ | $KGE_{qtot}$ | $KGE_{qbase}$ | $KGE_{mean}$ | $KGE_{qtot}$ | $KGE_{qbase}$ | |
| Lyne and Hollick - $KGE_{mean} = 0.4 \times KGE_{qtot} + 0.6 \times KGE_{qbase}$ | | | | | | | |
| Best compromise | $0.76_1$ | $0.80_7$ | $0.72_9$ | $0.70_9$ | $0.71_9$ | $0.70_2$ | 186 |
| 25 best compromises | $0.75_8 - 0.76_1$ | $0.78_1 - 0.82_0$ | $0.71_8 - 0.74_5$ | $0.69_7 - 0.71_6$ | $0.70_6 - 0.73_9$ | $0.67_3 - 0.72_1$ | 181 - 194 |
| Lyne and Hollick - $KGE_{mean} = 0.5 \times KGE_{qtot} + 0.5 \times KGE_{qbase}$ | | | | | | | |
| Best compromise | $0.77_2$ | $0.81_9$ | $0.72_5$ | $0.70_3$ | $0.71_4$ | $0.69_2$ | 184 |
| 25 best compromises | $0.77_0 - 0.77_2$ | $0.80_5 - 0.82_5$ | $0.71_8 - 0.73_7$ | $0.69_8 - 0.70_6$ | $0.70_4 - 0.71_5$ | $0.68_2 - 0.70_4$ | 181 - 188 |
| Lyne and Hollick - $KGE_{mean} = 0.6 \times KGE_{qtot} + 0.4 \times KGE_{qbase}$ | | | | | | | |
| Best compromise | $0.78_2$ | $0.82_5$ | $0.71_9$ | $0.70_2$ | $0.71_3$ | $0.69_4$ | 181 |
| 25 best compromises | $0.78_0 - 0.78_2$ | $0.81_4 - 0.83_5$ | $0.69_9 - 0.73_0$ | $0.69_5 - 0.70_8$ | $0.70_9 - 0.73_5$ | $0.64_6 - 0.70_1$ | 176 - 188 |
| Eckhardt - $KGE_{mean} = 0.4 \times KGE_{qtot} + 0.6 \times KGE_{qbase}$ | | | | | | | |
| Best compromise | $0.81_1$ | $0.87_0$ | $0.77_2$ | $0.68_6$ | $0.79_0$ | $0.61_7$ | 248 |
| 25 best compromises | $0.81_0 - 0.81_1$ | $0.86_7 - 0.87_2$ | $0.76_9 - 0.77_2$ | $0.68_0 - 0.68_6$ | $0.78_6 - 0.79_0$ | $0.60_8 - 0.61_8$ | 248 - 252 |
| Chapman- $KGE_{mean} = 0.4 \times KGE_{qtot} + 0.6 \times KGE_{qbase}$ | | | | | | | |
| Best compromise | $0.80_1$ | $0.87_0$ | $0.75_6$ | $0.69_5$ | $0.76_2$ | $0.65_0$ | 208 |
| 25 best compromises | $0.79_8 - 0.80_1$ | $0.86_9 - 0.87_5$ | $0.74_8 - 0.75_6$ | $0.68_9 - 0.69_9$ | $0.76_1 - 0.77_4$ | $0.63_5 - 0.65_4$ | 208-215 |

**Optimized parameters**

| | $T_M$ (°C) | $C_M$ (mm/°/d) | $TT_F$ (°C) | $F_T$ (d) | $t_{API}$ (d) | $f_{runoff}$ (-) | $sw_m$ (mm) | $f_{inf}$ (d⁻¹) |
|---|---|---|---|---|---|---|---|---|
| Lyne and Hollick - $KGEmean = 0.4 \times KGEqtot + 0.6 \times KGEqbase$ | | | | | | | | |
| Best compromise | 0.5 | 4.0 | -17.9 | 20.0 | 3.8 | 0.54 | 308 | 0.05 |
| 25 best compromises | 0.2 - 0.8 | 3.5 - 4.4 | -20.0 - -14.4 | 5.0 - 28.4 | 3.0 - 4.0 | 0.52 - 0.56 | 227 - 439 | 0.04 - 0.06 |
| Lyne and Hollick - $KGEmean = 0.5 \times KGEqtot + 0.5 \times KGEqbase$ | | | | | | | | |
| Best compromise | 1.0 | 4.5 | -19.0 | 28.4 | 3.0 | 0.54 | 300 | 0.05 |
| 25 best compromises | 0.6 - 1.0 | 4.2 - 4.5 | -19.0--14.8 | 13.7 - 28.4 | 3.0 - 3.3 | 0.51 - 0.56 | 252 - 366 | 0.05 - 0.06 |
| Lyne and Hollick - $KGEmean = 0.6 \times KGEqtot + 0.4 \times KGEqbase$ | | | | | | | | |
| Best compromise | 1.0 | 4.5 | -19.0 | 28.4 | 3.0 | 0.55 | 262 | 0.05 |
| 25 best compromises | 0.5 - 1 | 3.5 - 4.5 | -19.0 - -15.1 | 13.1-30.0 | 3.0 - 3.4 | 0.53 - 0.58 | 200 - 333 | 0.04 - 0.07 |
| Eckhardt - $KGEmean = 0.4 \times KGEqtot + 0.6 \times KGEqbase$ | | | | | | | | |
| Best compromise | 0.0 | 3.0 | -17.5 | 5.0 | 2.9 | 0.50 | 238 | 0.08 |
| 25 best compromises | -0.2 - 0.2 | 3.0 | -20.0 - -10.9 | 5.0 - 13.5 | 2.4 - 3.0 | 0.50 | 173 - 238 | 0.07 - 0.09 |
| Chapman - $KGEmean = 0.4 \times KGEqtot + 0.6 \times KGEqbase$ | | | | | | | | |
| Best compromise | -0.3 | 3.0 | -20.0 | 19.0 | 5.0 | 0.50 | 345 | 0.30 |
| 25 best compromises | -0.4 - -0.2 | 3.0 - 3.2 | -20.0 - -11.9 | 8.1 - 28.1 | 5.0 | 0.50 - 0.60 | 93 - 488 | 0.10 - 0.30 |

[Figure]

*Figure 1: Daily river flow from the downstream gauging station of W6 and baseflow estimated with the Lyne and Hollick filter (1979), the Eckhardt (2005) filter, and the Chapman (1991) filter*

[Figure]

*Figure 2: Monthly river flow from the downstream gauging station of W6 and baseflow estimated with the Lyne and Hollick filter (1979), the Eckhardt (2005) filter, and the Chapman (1991) filter*

[Figure]

*Figure 3: Monthly baseflow estimated with the Lyne and Hollick filter (1979), the Eckhardt (2005) filter, and the Chapman (1991) filter for the downstream gauging station of W6 and simulated potential GWR with the 25 best calibration parameter sets for each method*

**Table 3: Objective functions for the simulated outputs, for calibration and validation periods, and mean bias over the entire period of measurement**

| Gauging stations | | Calibation | | | Validation | | | Mean bias (all period - mm/month) | |
|---|---|---|---|---|---|---|---|---|---|
| Number | Measur. period | $KGE_{qtot}$ | $KGE_{qbase}$ | $KGE_{mean}$ | $KGE_{qtot}$ | $KGE_{qbase}$ | $KGE_{mean}$ | River flow | Pot. GWR |
| **W1*** 2 | 1980-2013 | 0.80 | 0.65 | 0.71 | 0.79 | 0.63 | 0.70 | 3 | -1 |
| **W2*** 5 | 1973-2017 | 0.76 | 0.56 | 0.64 | 0.75 | 0.53 | 0.62 | 5 | 1 |
| **W3** 14 | 1965-2017 | 0.76 | 0.64 | 0.69 | 0.79 | 0.57 | 0.66 | 5 | 0 |
| **W4*** 8 | 1961-2009 | 0.81 | 0.71 | 0.75 | 0.85 | 0.71 | 0.77 | -4 | 1 |
| **W5** 4 | 1961-2017 | 0.77 | 0.75 | 0.76 | 0.76 | 0.59 | 0.66 | -9 | -3 |
| **W6** 8 | 1961-2017 | 0.84 | 0.61 | 0.70 | 0.71 | 0.63 | 0.66 | -2 | -5 |
| **W7** 2 | 1993-2017 | 0.92 | 0.75 | 0.82 | 0.86 | 0.72 | 0.77 | 2 | 0 |
| **W8** 8 | 1961-2015 | 0.80 | 0.67 | 0.72 | 0.77 | 0.64 | 0.69 | -4 | -3 |

*\*The presented values are for the stations of which the watershed are completely located in Quebec*

**Table 4: Simulated runoff, actual evapotranspiration (AET), and potential groundwater recharge (GWR) and uncertainty for the study area between 1961 and 2017, in mm/year and in percentage for winter, spring, summer and fall, and for the eight watersheds (W1 to W8).**

| | Runoff | | | | | AET | | | | | Pot. GWR | | | | |
|---|---|---|---|---|---|---|---|---|---|---|---|---|---|---|---|
| | mm/yr | Win. | Spr. | Sum. | Fall | mm/yr | Win. | Spr. | Sum. | Fall | mm/yr | Win. | Spr. | Sum. | Fall |
| **W1*** | 368 ± 8 | 13% | 47% | 16% | 24% | 482 ± 5 | 2% | 30% | 48% | 20% | 109 ± 4 | 38% | 46% | 3% | 14% |
| **W2*** | 430 ± 9 | 12% | 48% | 16% | 24% | 498 ± 5 | 1% | 29% | 50% | 19% | 119 ± 5 | 36% | 45% | 4% | 15% |
| **W3** | 442 ± 10 | 12% | 48% | 17% | 24% | 507 ± 5 | 1% | 28% | 52% | 19% | 139 ± 5 | 35% | 44% | 4% | 17% |
| **W4*** | 485 ± 9 | 11% | 50% | 17% | 22% | 512 ± 4 | 1% | 25% | 55% | 18% | 147 ± 6 | 31% | 42% | 8% | 19% |
| **W5** | 438 ± 10 | 10% | 50% | 17% | 23% | 502 ± 5 | 1% | 26% | 54% | 19% | 144 ± 6 | 32% | 43% | 6% | 19% |
| **W6** | 465 ± 9 | 8% | 53% | 16% | 23% | 501 ± 4 | 1% | 25% | 56% | 18% | 151 ± 6 | 28% | 44% | 7% | 21% |
| **W7** | 453 ±8 | 6% | 56% | 15% | 23% | 512 ± 4 | 1% | 25% | 56% | 18% | 154 ± 5 | 26% | 46% | 8% | 20% |
| **W8** | 470 ± 10 | 7% | 53% | 18% | 23% | 494 ± 4 | 1% | 24% | 57% | 18% | 145 ± 6 | 27% | 42% | 10% | 21% |

*Part of the watershed is located in the USA - the presented values are for the Quebec part only*
*Winter: December, January, February; spring: March, April, May; summer: June, July, August; fall: September, October, November*

---

## Author Response (AR1)

We would like to thank the two referees for their comments and reviews aimed at improving our initial manuscript. Our answers to all the recommendations and how changes were introduced in the new version of the manuscript are described below. Therefore, all the cited lines in our answers hereafter refer to the lines of the revised manuscript. As well, the figures and tables that were modified in our revised manuscript are presented at the end of our answers.

**1. Answer to review of Anonymous Referee #1 (2021-04-02)**

**1.1. General comment**

**C1**: As a general comment, I found this manuscript too much focussed on the specific study area: no substantial new concepts, ideas, or methods. Moreover, I have some concerns on the way to present the results: I suggest the Authors, in case of resubmission to be much more concise.

Based on the remarks presented above, I suggest the editor to reconsider the manuscript after major revisions. I warmly suggest the Authors to "fly higher": the work done is a good basis for a scientific paper to be published on HESS, but some hints are not adequately developed to be interesting for a wider audience.

**A1**: We agree that the results needed to be presented in a broader, less local-context way and we have included major modifications to position the paper within the context of cold and humid climate research on groundwater recharge (GWR) and the changes in past GWR associated to the changing climatic conditions of the last decades. To reflect this, the title was modified to "Simulation of long-term spatiotemporal variations in regional-scale groundwater recharge: Contributions of a water budget approach in cold and humid climates". The introduction (L37-46, L57-68, L69-80) was modified to position the study within the recent research on GWR in cold and humid climates, such as Aygün et al., 2020; Grinevskiy et al., 2021; Kløve et al., 2017; and Nygren et al., 2020 (complete references in our revised manuscript).

The problem statement was also rephrased into (L81-82): "The objective of this study was to demonstrate the relevance of using a water budget model to understand long-term transient and regional-scale GWR in cold and humid climates where groundwater observations are scarce."

As well, sub-sections 5.1 and 5.2 were modified and rewritten to compare the regional-scale results to GWR estimates obtained in likewise cold and humid climates such as in Scandinavia and northern Russia and to include a comparison of the GWR trends within the warming climate observed in these regions since the middle of the 20th century.

**1.2. Main comments**

**C2**: Lines 172-175. The Authors mention three basic hypothesis. I have serious concerns on the third one: the watershed response time is shorter than one month, thus compensating for the absence of water routing. This is a very strong hypothesis, as it allows neglecting transient dynamics of the aquifer. Before proceeding with any computation, Author should demonstrate and convince the reader that this a reliable assumption.

**A2:** Monthly time steps are frequently used in groundwater flow modelling and are considered a reasonable timescale for groundwater flow processes in cold and humid climates (Dripps and Bradbury, 2007; Guay et al., 2013). To avoid misleading the reader, L161-168 were rephrased: "The HydroBudget model is calibrated on superficial watersheds, based on the hypotheses that: 1) surface watersheds match hydrogeological watersheds, 2) the rivers drain unconfined aquifers. Under these conditions, for any given watershed, potential GWR should be similar to river baseflow at the outlet, and the sum of runoff and potential GWR should be equal to the total flow at the outlet. Dripps and Bradbury (2007), and Guay et al. (2013) have shown that results of GWR water budget models (SWB and HELP, respectively), compared better to groundwater flow models (GFLOW and CATHY, respectively) at the monthly time step than at a daily time step. Thus, runoff, AET, and potential GWR were computed with HB with a daily time step, and the results were aggregated on monthly time steps. Calibration and validation of the model and analysis of the results were all performed on the monthly aggregated results."

**C3**: The model accounts for 8 parameters to be calibrated. It seems to me that some of them are set of parameters. For example the Runoff factor (as explained by the Authors "Partitioning between runoff computed with the RCN method and infiltration into the soil reservoir") do depend also on the land cover or not? in the first case, you are calibrating each runoff factor for each soil. Am I wrong? Maybe I do not understand correctly.

**A3:** For clarification purpose, the following sentence was added L141-142: "These parameters are held constant for all the grid cells and through time (Table 2)." As well, the description of  $f_{runoff}$  parameter in the Table 2 of our revised manuscript was modified for: "Correction factor for runoff computed with the RCN method for the partitioning between runoff and infiltration into the soil reservoir".

**C4**: The coupling of time steps (daily time step for soil modelling and monthly time step for GWR) is not clear to me. Please, give more details on that.

**A4:** Clarification was added about the use of the daily time-steps for the model calculations aggregated into monthly time-step for the calibration and validation and the result analysis (L166-168 – presented in our answer A2): "Thus, runoff, AET, and potential GWR were computed with HB with a daily time step, and the results were aggregated on monthly time steps. Calibration and validation of the model and analysis of the results were all performed on the monthly aggregated results."

**C5:** The targets for calibration are both the total surface flow and the baseflow. However, the first is observed, whereas the second is estimated (through the Lyne and Hollick filter). In my opinion this is a weak point of the whole calibration procedure: generally speaking I do not like to calibrate a model using output of another model. Even if they match each others, what can I say on the reliability of both? The Authors should at least convince the reader on the reliability of the baseflow estimates.

**A5:** We agree with the referee that baseflow estimates can be seen as a sensitive point of our calibration procedure. To address this issue, the influence of the baseflow separation filter on the calibration and simulation results was tested with three filters on the group of gauging stations in W6. The text now reads as follows (L170-174): "Baseflows were estimated from the flow rate time series following the proposition of Ladson et al. (2013) with the Lyne and Hollick filter (Lyne and Hollick, 1979) and using a stochastic calibration and 30 filter passes. To compare the effects of the baseflow filters on GWR simulation, baseflows were also estimated with the Eckhardt (2005) and Chapman (1991) filters for the gauging stations of W6."

To reflect this, the supplementary Table A. 1 of our manuscript was modified as presented hereafter.

As well, sentence L173-175 was added to acknowledge the fact that baseflows are an acceptable proxy for groundwater recharge in the hydrogeological context of southern Quebec: "As aquifers generally discharge into superficial water bodies in southern Quebec, baseflows estimated with recursive filters are considered an acceptable proxy of GWR dynamics."

The results of the calibration of the gauging stations of W6 with the three baseflow filters are presented in L237-240: "As the absolute value of baseflow changed with the filter method, the interannual GWR varied in consequence. The calibrated parameters with the different baseflow estimates slightly varied but remained in the parameter ranges described in Table 2. The objective functions were similar, between 0.7 and 0.8, except for  $KGE_{qbase}$  in validation, which was slightly lower with the Eckhardt (2005) and Chapman (1991) filters than for Lyne and Hollick (1979) filter."

The discussion about the importance of using baseflows was modified (L472-480): "The simulated transient GWR was calibrated with baseflows computed using regressive filters from river flow rate time series. Although Partington et al. (2012) showed that the association between baseflows and GWR is not always satisfactory and different baseflow separation methods can lead to differences in volumes and timing (Gonzales et al., 2009; Zhang et al., 2017), baseflows are generally considered an acceptable proxy for GWR in cold and humid climates (Chemingui et al., 2015; Dierauer et al., 2018; Rivard et al., 2009). They are widely used for the calibration of GWR simulations (Batelaan and De Smedt, 2007; Croteau et al., 2010; Dripps and Bradbury, 2007; Gagné et al., 2018; Rivard et al., 2013). In this work, using baseflow estimated with the Lyne and Hollick (1979), Eckhardt (2005), and Chapman (1991) filters influenced the proportion of baseflow in total river flow, and consequently led to variations in simulated potential GWR (Supplementary Table A. 1). However, the quality of the simulations was comparable due to parameter variations that remained in the possible range of the parameters (Table 2)."

**C6**: The previous one is a very important point, also because the objective function (eq. 1) is a linear combination of two different metrics, one referring to the total flows, the second one only to the baseflows. In my opinion, dependence of the calibration on the weights adopted to define the objective function should be explored much more in details. The explanation you gave for your choice (lines 203-205) is too simplistic.

**A6**: This is a valid observation, and we thank the referee for underlying the importance of these weights. To clarify this, different  $KGE_{mean}$  computation weight were tested for the group of gauging stations of W6 (L199-202): "The weights attributed to each objective function in  $KGE_{mean}$  were arbitrarily chosen to select the calibrated parameter set that maximizes the reproduction quality of the baseflows, considered to be the proxy for GWR ( $KGE_{qbase}$ ), without losing the benefits of the multi-objective optimization (Supplementary Table A. 1). The impact of the chosen weights was tested for the gauging stations of W6."

The results are presented as follows in L235-237: "The objective functions, the calibrated parameters, and the interannual potential GWR values showed very limited sensitivity to the weights chosen for the computation of the *KGEmean* (supplementary Table A. 1)."

**C7:** I found the sensitivity analysis performed on the W6 group of gauging stations very interesting and potentially the core business of the paper (which otherwise it is only a modelling study, important for Quebec, but not interesting for the international reader of HESS). Why the Authors decided to carry out the sensitivity analysis only on one group of gauging stations. In my opinion, it could be much more interesting to perform the same analysis to several groups of gauging stations to explore the possible differences among watershed and the dependences on climate forcing, soil, land cover etc. In other words, has the ranking proposed for W6 a general validity? why?

**A7**: We thank the referee for this important comment. As suggested, the sensitivity analysis was carried out for the eight groups of gauging stations as explain L207-209: "Using the R package *Sensitivity* (https://CRAN.R-project.org/package=sensitivity), the Morris global sensitivity approach (Morris, 1991), enhanced by Campolongo et al. (2007), was performed for the eight groups of gauging stations and for the two objective functions, *KGE*qtot and *KGE*qbase."

The results are presented in Fig. 4, which was added to the revised manuscript.

The presentation of the sensitivity analysis results was modified for (L219-228): "The model sensitivity was obtained with 60 repetitions of the design for the eight groups of gauging stations, representing 540 model calls for each group. The relative sensitivity of the model varied markedly between the groups of gauging stations for the simulated river flow rates (Fig. 4a) but appears to be more constant for the simulated potential GWR (Fig. 4b). Flow rates were mostly sensitive to the snow-related parameters ( $T_M$  and  $C_M$ ), except for the western watersheds where the  $f_{runoff}$  was more important. The simulated potential GWR was most sensitive to  $f_{runoff}$  and least sensitive to snowmelt parameters ( $T_M$  and  $C_M$ ) for all the watersheds. The ranking from the second to the fifth-highest sensitivity of potential GWR only slightly varied between groups

of gauging stations. River flow rates and potential GWR clearly showed limited sensitivity to the soil freezing time ( $F_T$ ) with only a slightly higher sensitivity for the eastern watersheds W7 and W8. Overall, the potential GWR was more sensitive to parameter variations than river flow since all the  $\mu^*$  for the river flow were lower by a factor 2 to 10 than for the potential GWR (values not presented here).

The contribution of the sensitivity analysis was discussed L443-449: "The results showed that the simulated potential GWR was sensitive to  $TT_F$ , while both flow rates and potential GWR had limited sensitivity to  $F_T$  (Fig. 4). The colder watersheds were more sensitive to these parameters while the simulations of river flows in the warmer watersheds were less sensitive to the snow-related parameters. These results underline the importance of including soil freezing in GWR modeling for cold regions, as did other studies in cold and humid climates (Grinevskiy et al., 2021; Nemri and Kinnard, 2020; Okkonen and Kløve, 2011). This work pinpoints the need for more research on winter recharge to better understand the processes involved."

A related addition was also included in the conclusion L507-509: "Another outcome of this work was to show that the model sensitivity to its parameters is correlated to the average air temperature of the watershed, making the simulated water budget in watersheds with lower temperature more sensitive to snow-related parameters than the warmer watersheds."

**C8**: Presentations of the results are sometimes not easily readable. For example, in section 4.3 one can find several information already shown in figure 6 and table 4). I think that the description of the results (sections 4.2, 4.3, 4.4) can be much simplified. On the contrary, results on the temporal evolution (analysis of trends) deserves for much more attention and a dedicated picture to presents results.

**A8**: As suggested, the descriptions of the results in the sections 4.2, 4.3, and 4.4 have been significantly simplified (cf. L254-262, L274-278, L279-283, L295-301), Fig. 6 of the initial manuscript was withdrawn, and the description of the changes in the past water budget was discussed and associated to the changes in the precipitation and temperatures since 1961. Consequently, **Fig. 8** of the manuscript was modified as presented hereafter. The comparison of the 19-year periods is presented L302-308: "The simulation period was split into three 19-year periods (1961-1979, 1980-1998 and 1999-2017) to look for significant changes in input variables (precipitation and temperature) and simulated variables (Tukey test, p < 0.05) (Fig. 8). Significant changes were found in the input variables between the 1961-1979 and the 1999-2017 periods, with an increase of temperature for all watersheds and increase of precipitation in five watersheds. These changes produced a significant increase of AET for all watersheds except W1, and an increase in runoff for three watersheds (W3, W6 and W8). However, these changes in precipitation and temperature did not impact the annual potential GWR for which significant increases were simulated between the 1961-1979 and the 1961-1979 and the 1961-1979 and the 1961-1979 and the 1961-1979.

The discussion about the trends was also revised as follows (L309-317): "When considering the entire period (1961-2017), VI significantly increased (Mann-Kendall test, p < 0.05) annually for all watersheds except W5 and W7 and during winter (December to February) and summer (June to August) for all

watersheds except W1 and W7 during summer (supplementary Table A. 2). Significant annual and seasonal temperature increases were also observed for all watersheds and for all seasons except W5 and W6 during spring. Similar to VI, runoff significantly increased annually for all watersheds except W5 and W7 and during winter and summer for all watersheds except W1 during summer. Significant increases in AET were also simulated for annual values and during winter and spring, and unlike temperature, summer and fall AET showed no significant trend except for W6 and W8 during summer. Significant increases in potential GWR were simulated only for winter values for all watersheds except W1. Interestingly, although increasing runoff and AET were observed throughout the simulation period (1961-2017), no decreasing trends were simulated in potential GWR."

These results are discussed in the fully rewritten section 5.2, L372-383: "Warming temperature were recorded in the cold and humid climates of the northern hemisphere since the middle of the 20th century. Mean annual temperature has increased by +2°C to +3°C between 1954 and 2003, and this increase has been more marked between December and February (up to +4°C for the same period) (Arctic Climate Impact Assessment, 2004). In eastern Canada, temperature warming up to +1°C to +3°C has been identified for the 1948-2016 period (Vincent et al., 2018), leading to important changes in the seasonal distribution, and particularly to an increase in winter precipitation and a decrease of accumulated snow during the cold season (Kong and Wang, 2017). This work has shown that although the simulated potential GWR was not impacted by the long-term increases in precipitation or temperature on an annual time-step for the 1961-2017 period, these changes produced significant increases in annual runoff and AET and may not have been sufficient yet to propagate to potential GWR. As observed with past data for winter trends in potential GWR, under changing climate conditions, seasonal and monthly GWR could increase during periods with more available liquid water and low AET rates (late fall, winter, early spring). Inversely, GWR could decrease for periods of enhanced AET rates (warmer temperature) such as late spring, summer, and early fall."

This is also summarized in the conclusion L515-519: "Interestingly, the long simulation period made it possible to identify a significant increasing trend of GWR during winter and no decreasing trends during summer, despite warmer temperatures. This is most likely because increases in AET are compensated by increases in precipitation. In contrast to previous studies of past GWR trends in cold and humid climates, this work has shown that changes in past climatic conditions have not (yet) produced significant changes in annual potential GWR but have impacted runoff and AET."

**C9**: The entire discussion on the temporal patterns of groundwater recharge relates the time variation of meteorological forcing to discharge, neglecting possible changes in the land cover. I do not know if such an assumption is correct, however it should be explicitly stated

**A9**: This is a very good point that needed to be mentioned. The following sentence was added to explicitly state that the simulation was performed assuming that the land use did not change L168-169: "As well, the

calibration and validation of the model were performed under the hypothesis that no major land use change occurred during the simulation period (*i.e.* land use was held constant)."

**C10**: Results of figure 6 are surprising: proportion between runoff, AET, GWR does not depend on the watershed (differences among watersheds are in the order of 1-2%). Figure 5(b) shows different patterns: for example over W2 GWR/P spans between 0.1 and 0.15 for most of the grids; on the contrary, over W4, it seems that GWR/P is >0.3 for half of the cells. Maybe I missed something, but it does not seem to me that Fig 5 and Fig 6 are coherent.

**A10: [The Fig. 6 of the initial manuscript has been withdrawn]**

The results were double-checked and there was no mistake. Considering the spatial variations in precipitation in the study area (952 mm/yr in W1, 1 039 mm/yr in W2, and 1 123 mm/yr in W4; table 1; Fig. 2 of the manuscript), variations are found between absolute values of potential GWR (109 mm/yr in W1, 119 mm/yr in W2, and 147 mm/yr in W4; Table 4 of the manuscript). The ratios of average potential GWR/average precipitation for these three watersheds are 0.114, 0.115, and 0.131, respectively. When computed from the spatially-distributed GWR/P ratio presented in Fig. 6b of the revised manuscript, the averaged ratios for the same three watersheds are 0.114, 0.113, and 0.131, respectively. We believe that the differences in precipitation explains that the two figures from the initial manuscript were coherent although it seems that the GWR rates would be higher in W4.

**1.3. Minor comments**

**mC1: Line 102-109 All the information given here are summarized in Table 1. This lines appear to me not useful.**

We have modified L90-97 as follows: "The case study area is located in the province of Quebec (humid and cold climate), between the St. Lawrence River and the Canada-USA border, and between the Quebec-Ontario border and Quebec City (35 800 km2) (Fig. 1). It is comprised of the watersheds of eight main tributaries of the St. Lawrence River (numbered 1 to 8 from west to east) (Table 1). Watersheds W1 (Châteaugay River), W2 (Richelieu River), and W4 (Saint-François River) are partially located in the USA (42%, 83%, and 15% of their total areas respectively). Topography is flat with low elevation areas close to the St. Lawrence River and higher elevations in the Appalachian Mountain range, associated with steeper slopes. Land cover includes agriculture, forest, wetlands, urban uses, and surface water (Fig. 2a). Agriculture dominates in the watersheds located in the St. Lawrence Platform, while forest occupies most of the Appalachian watersheds."

**mC2**: Line 112. In my opinion it would be better to add also the mean bias as metrics of the goodness of interpolation, to avoid systematic under or overestimation.

We have modified L99-102 as follows : "The high density of measurements during this period generated minimal error on the interpolated data, with a root mean square error (RMSE) of 3 mm/d for precipitation (mean bias of +0.1 mm/d), 2.5°C for minimal temperature (mean bias of -0.5°C), and 1.5°C for maximal temperature (mean bias of +0.1°C; Bergeron, 2016)."

**mC3**: Figure 1. In my opinion the map in the middle is not useful: I suggest to eliminate it, retaining the location map and the watersheds map.

Fig. 1 of the revised manuscript was modified as presented hereafter.

**mC4**: Figure 4. As GWR is your main output, I would present several graphs as the panel (b) for several stations (one as an example is not useful)

[Fig. 4 of the initial manuscript is the Fig. 5 in the revised manuscript]

Fig. 5 was modified as presented hereafter.

The associated paragraph L254-262 was rephrased as follows: "The calibrated HB model was used to simulate potential GWR for the entire study area on a 500 m x 500 m grid for the 1961-2017 period. Examples of the simulated monthly baseflows (Lyne and Hollick filter) and potential GWR are illustrated in Fig. 5 for the downstream stations in W3, W7, and W8. The simulated potential GWR compared favorably with baseflows estimated using the Lyne and Hollick (1979) digital filter. Maximum values were reached simultaneously in April, during the spring month(s) of maximum VI. A second baseflow and GWR peak was observed and simulated in November-December of most years. Lowest values were reached in July to September (high AET rates) and February (minimum VI). Similar matching results in timing and amplitude were obtained for the river flow (not presented here). Simulated AET was null in the winter until the spring thaw, after which it quickly reached its highest value (> 100 mm/month) in July and decreased at the end of August, reaching null values again in November. Comparable results were obtained for the other gauging stations (not shown)."

**2. Answer to review of Anonymous Referee #2 (2021-05-17)**

**2.1. Main comments**

**C11**: In response to reviewer #1, comment 5 [main comment C6 – section 1.2], you provide a helpful table of how calibration results change with different weights in the objective function. You say that the calibration is "moderately sensitive to the weights" – but what implication does this have for the results? Do these results all fall within the sensitivity analyses? The key question here is does changing the objective function change the GWR estimates and trends? Would your interpretations and conclusions change?

**A11**: As presented in answer A6 (section 1.2) and in the modified supplementary **Table A. 1** of the revised manuscript, we compared the influence of three pair of weights on the simulation results. Objectives functions, calibration parameters, and GWR estimates showed very limited sensitivity to the chosen weights, as concluded L235-237: "The objective functions, the calibrated parameters, and the interannual potential GWR values showed very limited sensitivity to the weights chosen for the computation of the *KGEmean* (supplementary Table A. 1)."

Therefore, the chosen weights did not impact the interpretation and conclusion of the study.

**C12**: Further to #1 [previous C11], and in line with reviewer #1's comment 4 [main comment C5 – section 1.2], what effect does the selection of baseflow separation method have on the results? As you point out in the discussion and in the response to reviewer #1 there can be significant variance in baseflow estimates between different methods. If you selected a different method, would the interpretation change? I acknowledge that these analyses may be time consuming, but I think they speak to the robustness of the approach. These baseflow estimates come with such high uncertainty that I feel they should almost be treated as another parameter – how would your results change if baseflow varied?

**A12**: Please, refer to our answer A5 in section 1.2 where we detail the modifications to the initial manuscript following the calibrations performed with the Lyne and Hollick (1979), Eckhardt (2005), and Chapman (1991) filters to estimate the baseflows. As mentioned in our answer A5 in section 1.2, results are presented in **Table A. 1** and L237-240 and are discussed L472-480.

**C13**: I really appreciate how this work can take trends in baseflow (estimated from measurements at sparse gage stations) and use that to estimate the spatial distribution of potential groundwater recharge across a watershed. However, I think the manuscript would benefit from a more explicit discussion of the uncertainties that propagate through the workflow. Again, this kind of analysis and discussion really helps to make the results more robust and applicable.

**A13**: As suggested by the first Referee to add the mean bias in the input data (precipitation and temperature, minor comments mC2 – section 1.3), the mean bias for simulated river flow and GWR was

added in **Table 3** of the revised manuscript (presented hereafter) to compare the error in the simulated variables to the error in the input data.

The uncertainty was quantified running the model with the 100 best sets of regionalized parameters and calculating the standard deviation between the 100 model runs for each month, as presented L202-204: "The 100 best compromises of each group of gauging stations were used to produce the 100 best regionalized parameter sets and the HB model was run with these parameters, estimating uncertainty from the standard deviation."

The uncertainty for the simulated runoff, AET, and GWR was added in the **Table 4** of the revised manuscript (presented hereafter).

The results are presented L249-251: "The mean bias on simulated river flow rates and potential GWR varied between -9 mm/month and 5 mm/month (Table 3). The uncertainty computed with the 100 best regionalized parameter sets was  $\leq$  10 mm/yr for the three simulated variables (Table 4)."

This was added in the discussion L439-440: "Furthermore, the uncertainty analysis showed that the calibration method provides an interestingly small uncertainty for the simulated variables, and a mean bias similar to that of the input data."

These results are also included in the conclusion L503-504: "With the model simultaneously calibrated on 51 gauging stations, GWR was simulated between 1961 and 2017 at the regional scale (36,000 km2) with very little uncertainty (<10 mm/yr)."

**C14**: You compare results from this work to results from previous studies – a lot of them spatially-distributed numerical models. Can you provide some idea of how the spatial distribution of GWR compares between them, as opposed to just the mean/variance/ranges? Are you picking up the same high/low recharge patterns? The same temporal distributions? This is hinted in section 5.1 but I Think a more robust comparison would improve the manuscript.

**A14**: This particular point was not developed in our revised manuscript since the comparison with local studies was shortened and the comparison with other studies in cold and humid climatic environments was developed (L353-359 and L360-369). As well, the re-written section 5.2 of the revised manuscript now compares the evolution of the GWR simulated over the study area for the last decades to observations of trends in GWR in cold and humid climates made in other studies (L384-394).

**2.2. Minor comments**

**mC5**: *Line 60-61: It is bizarre that the reference that "acknowledges the lack of representiveness of the daily results…." Is from 2007, yet the reference for the model they are acknowledging is from 2010.*

Indeed, the paper from Dripps and Bradburry (2007 – full reference in our manuscript) was the reference for the SWB model, therefore, the citation of Westenbroek et al. (2010) was removed. L64-68 were rephrased: "For example, similar monthly and annual GWR estimates were obtained in cold and humid climates with the Soil Water Balance model (SWB) and the GFLOW analytical model (Hunt et al., 1998) by Dripps and Bradburry (2007) in Wisconsin (U.S.A) or with the HELP water budget model (Schroeder et al., 1994) and the fully coupled CATHY model (Comporese et al., 2010) by Guay et al. (2013) in Quebec (Canada)."

mC6: Line 80-81: I think this is needed in a lot of places, not just southern QC!

The paragraph has been removed.

**mC7**: *Line 304-305* [L289-290 of the revised manuscript]: So it is associated more with the precip trends than with the soil type?

Clayey areas are mainly located in the St. Lawrence Lowlands, the flattest of the study area, that also happen to receive the less rainfall (mainly < 1 000 mm/yr) in the study area. The combination of the three (precipitation distribution, soil type, and flat topography) explains that besides having the smallest potential GWR rates of the study area, clayey areas are also associated with the smallest runoff rates and AET rates. The text was not modified.

**mC8**: Section 4.4; 1st para: This is a rough paragraph to read. I think a table or graph would be more appropriate.

The first paragraph of section 4.4 L295-301 was rephrased as follows: "HydroBudget simulated the temporal evolution of the water budget in the study area from 1961 to 2017, thus producing an exceptionally long simulated time series of runoff, AET, and potential GWR for the area (Fig. 8). The effect of interannual variability in precipitation appeared clearly in the simulated runoff with low runoff rates (< 350 mm/yr) produced during the driest year and high runoff rates (> 550 mm/yr) produced during the wettest year (Fig. 8a, c). The simulated AET varied less than runoff, mainly between 450 mm/yr and 560 mm/yr (Fig. 8d). Interannual variations of potential GWR were relatively high, comprised between 90 mm/yr and 200 mm/yr, and seemed more influenced by precipitation than by temperature variations (Fig. 8e)."

As well, to make the presentation of the results clearer, Fig. 8 of the original manuscript was modified as presented in **Fig. 8** hereafter.

**mC9: Line 343-344: "somewhat higher" is almost double.**

The expression "somewhat higher" was changed for "much higher" in L328-331: " For example, Chemingui et al. (2015) used the CATHY integrated hydrological model in W1 to estimate GWR to be 200 mm/yr. Although this value is much higher than the one obtained using HB in the same area (109 mm/yr), the resulting preferential recharge areas located close to the Canada-USA border are similar with both approaches (i.e., 70 mm/yr to 250 mm/yr in Chemingui et al. (2015) and 70 mm/yr to 280 mm/yr with HB)."

**mC10**: *Line 381-382: awkward sentence structure. Maybe provide range in main sentence text (between 89 and 198) and then average in brackets?*

The sentence was removed from the paragraph L340-347 in section 5.1.

**mC11**: Line 387: It is hard to support that this is the novel contribution when you follow it up with the fact that it is supported by a lot of previous literature. I would consider rephrasing this to highlight the specific contribution beyond what the existing literature provides.

The paragraph was modified for (L353-359): "This spatiotemporal link between GWR and precipitation and temperature patterns is coherent with that reported in other studies. For example, Dierauer et al. (2018) used river flow observations of 63 streams in the Rocky Mountains (Canada and USA, snow-dominated hydrology), to link winter conditions to GWR. The proportion of total precipitation occurring in winter and winter temperature were identified as important variables for the variability of winter and summer low flows. The seasonality of GWR is well documented in the humid conditions of northern Europe as well, with an increase of the importance of winter events (rain-dominated GWR) at the expense of the spring snowmelt (snow-dominated GWR) with the increase of average temperature from north to south (Kløve et al., 2017; Nygren et al., 2020)."

**mC12**: Line 399: This is a regional contribution. While significant to those interested in this region, authors should try to focus on the broader applicability and contributions.

Section 5.2 was entirely rewritten and L372-383, L384-394, and L395-398 were rephrased into: "Warming temperature were recorded in the cold and humid climates of the northern hemisphere since the middle of the 20th century. Mean annual temperature has increased by +2°C to +3°C between 1954 and 2003, and this increase has been more marked between December and February (up to +4°C for the same period) (Arctic Climate Impact Assessment, 2004). In eastern Canada, temperature warming up to +1°C to +3°C

has been identified for the 1948-2016 period (Vincent et al., 2018), leading to important changes in the seasonal distribution, and particularly to an increase in winter precipitation and a decrease of accumulated snow during the cold season (Kong and Wang, 2017). This work has shown that although the simulated potential GWR was not impacted by the long-term increases in precipitation or temperature on an annual time-step for the 1961-2017 period, these changes produced significant increases in annual runoff and AET and may not have been sufficient yet to propagate to potential GWR. As observed with past data for winter trends in potential GWR, under changing climate conditions, seasonal and monthly GWR could increase during periods with more available liquid water and low AET rates (late fall, winter, early spring). Inversely, GWR could decrease for periods of enhanced AET rates (warmer temperature) such as late spring, summer, and early fall.

In Canada, other authors have looked for trends in observed time series of groundwater levels for the snowdominated region of British-Columbia (Allen et al., 2014) or across the country (Rivard et al., 2009), but each time, no coherent pattern could be identified. This could be due to short time series or insufficient data coverage. In western Russia, Grinevskiy et al. (2021) simulated an increase of GWR between the 1965-1988 and 1989-2018 periods with a surface energy balance coupled to the Hydrus unsaturated zone model. The GWR increase was due to warmer winter leading to a decrease of soil frost depth and an increase of the winter soil-water storage. Interestingly, the warming temperatures did not produce AET increases which was interpreted to be due to a sharp decrease in wind speed. Inversely, Nygren et al. (2020) identified a significant decrease in groundwater depth time series across Finland and Sweden between the 1980–1989 and 2001–2010 periods. These authors showed that the deeper groundwater levels were linked to decreasing GWR pattern associated with shorter snowmelt periods and longer periods of intensive evapotranspiration. In comparison, the temperature increase observed in southern Quebec over the 1961-2017 period was probably not large enough to produce such a change.

An outcome of this work was to show that the regional long-term potential GWR simulations with a water budget allowed identifying the impacts of a changing climate on the past GWR conditions by positioning the trends (or absence of significant trends) in the globally changing hydrologic dynamic. As well, it reminded the high responsivity of the hydrologic dynamic of the cold months to the climate changes in the cold and humid climates."

**mC13**: Line 439: I generally try to avoid using unquantifiable terms like "good" – how else can you describe this that can be backed up by the results?

We agree with the Referee and the term "good" was changed for "satisfying" in the sentence L436-438: "Although the model does not compute water routing, groundwater-surface water feedback, or evapotranspiration from groundwater, and produces potential GWR, the satisfying simulation results found in this work justify the water budget calculation scheme used in HB, resulting in an easy-to-use and computationally efficient model."

**mC14: Line 490: 'work' instead of 'word'**

The term "word" was changed for "work" in the sentence L487-489: "An original contribution of this work was to show that the tested calibration method combined with a regionalized parameter set offers a very acceptable solution that is highly reproducible and could be applied in less monitored regions."

**mC15**: Line 515: Could this have been interpreted from trends in baseflow? Or does this work provide additional info that goes beyond that provided by baseflow results alone?**

This work provided more information on trends in GWR that an analysis of trends in baseflow for several reasons: 1) GWR is simulated continuously over the study area for the 1961-2017 period while baseflows estimated from the river flow time series are only available for periods of time (gaps in the time series and abandonment of gauging stations), i.e. Rivard et al (2009 – reference in the manuscript) did not find any consistent trend in Eastern Canada based on baseflow analysis. As well, computing trends on the simulated runoff and AET allowed understanding better how the trends in the climate data (increase of precipitation and warming temperature) translated into the regional hydrological dynamic (increase of runoff and AET, but not GWR – please refer to the re-written section 5.2 presented in mC12 here above). 2) River flow measurements during winter (low flow period) are highly uncertain due to ice cover and ice flowing, therefore analysing trends from the raw data only during these periods could be misleading. Conclusions of this work for future groundwater recharge assessment were added L448-449: "This work pinpoints the need for more research on winter recharge to better understand the processes involved." and L397-398: "As well, it reminded the high responsivity of the hydrologic dynamic of the cold months to the climate changes in the cold and humid climates."

**3. Figures and tables**

---

## Author Response (AR2)

Dear Referees, dear Editor,

We would like to thank you for your feedback and final suggestions of improvement for our work. All the propositions of modifications presented hereafter, and based on the Referee #3 comments, will be added to the precedent version of our manuscript.

*C1: The description of the HydroBudget area is still too vague. I could not rebuild the model with a scripting language if I wanted to from the description and the schematic figure alone. So, I urge the authors to also include (can be done in an appendix) the equations behind every reservoir's states and fluxes, particularly how the fluxes depend on the states. These may be time-explicit difference equations if these are being used, but they should be in to understand what is going on.*

A1: We would like to thank the Referee for pointing out that the HydroBudget description needed to be more precise. As suggested, we will include the equations in Appendix A.1, as shown at the end of this document (these equations are also found in the HydroBudget User Guide – Dubois et al., 2021).

L137-144 will be rephrased into: "HydroBudget (HB) is a spatially-distributed GWR model that computes a superficial water budget on grid cells of regional-scale watersheds with outputs aggregated into monthly time steps. The model uses commonly available meteorological data (daily precipitation and temperature, spatialized if possible) and spatially-distributed data (pedology, land cover, and slopes). It is based on simplified process representations and is driven by eight parameters that need to be calibrated. These parameters are uniform over the grid and held constant through time (**Erreur ! Source du renvoi introuvable.**). Coded in R, HB uses a conceptual lumped reservoir to compute the soil water budget on a daily time step (Appendix A.1). For each grid cell and each time step, precipitation is divided between runoff (R), evapotranspiration (ET), and infiltration that could reach the saturated zone (potential GWR), with a monthly time step (Erreur ! Source du renvoi introuvable.; Dubois et al., 2021)."

*C2: I assume that if the authors state (line 141) that "these parameters are held "constant" for all grids" they mean that they are "uniform" over the grid (the same value everywhere)? Please restate!*

A2: We agree that this sentence was not clear. It was rephrased L141-142 in the modified paragraph presented in A1: "These parameters are uniform over the grid and held constant through time (Table 2)."

As well, it appears as a footnote in Table 2.

*C3: It is not clear how the cell-specific land properties "pedology, land cover and slopes" come in to make it a distributed model. Are these properties only used in the CN method to calculate the discharge proportion? This would also become much clearer if we can see the equations and the equations denote which parameters are uniform and which location-dependent. Also, is the maximum storage capacity of the soil dependent on pedology or uniform across the grid as well?*

A3: We understand that the information about HydroBudget parameters was not precise enough and we thank the Referee for pointing it out. The Referee is correct, the spatialized pedology, land cover, and slopes are only used in the RCN computation. We followed the Referee's suggestion and highlighted the calibration parameters that are uniform over the grid and held constant in time in the new Appendix A1, and these include the $sw_m$ parameter (maximum storage capacity).

To clarify this point, L149-150 have been rephrased as follows: "Runoff is calculated using the runoff curve number (RCN) method (USDA-NRCS, 2004; 2007) on a cell-by-cell basis (two parameters, $t_{API}$ and $f_{runoff}$), similar to what is done in the SWAT model (Arnold et al., 2012; Neitsch et al., 2002). The RCN is attributed per cell based on its pedology, land cover, and slope following the USDA-NRCS method adapted for the Quebec environment (Dubois et al., 2021; Monfet, 1979)."

C4: *The calibration strategy is still not clear. Do you calibrate a separate set of parameters (8 in total) per sub-catchment (grouped discharge stations) or identify single sets for the entire catchments using a weighted criterion from sub-criterions of the different grouped stations? An equation of the criterion could help clarify this.*

A4: We think that it is a very good idea to be more precise about the calibration strategy, by adding more information about the equations of the objective functions and the regionalized parameters. We thank the Referee for suggesting it. To do so, L178-204 have been modified as follows:

[revised manuscript text omitted]

C5: *It is not clear how the calibrated parameters are "averaged" or "regionalized" and downscaled. What is the use of that if all grids get the same parameter values? Or is it downscaling in time? Please extent on this and use equations if this is possible.*

A5: We understand the Referee's concern, and we believe that the modified L178-204 presented in our previous answer resolves that question.

C6: *It the code of HydroBudget is open source, then I would urge the authors to place it somewhere where it can be downloaded directly, such as on GitHub.*

A6: We agree with the Referee that the code needs to be available for direct download. It will be deposited on a Dataverse linked to UQAM (https://dataverse.scholarsportal.info/dataverse/sp?q=). The code will be available to download with an application example on the smallest river watershed of our study area and an updated version of the User Guide.

The precise Internet address and the DOI of the documents will be available soon and the information will be added in the section 8 Code availability of the next version of the paper.

**Appendix A.1: Equations used in the HydroBudget model (adapted from Dubois et al., 2021 – the eight calibration parameters, uniform over the grid and constant through time, are identified with bold characters)**

*Degree-days snowmelt model*

Determining if the temperatures generates snowfall
If $T_t \leq 0$

$\qquad$ Then $snowfall_t = P_{TOTt}$ $\qquad\qquad\qquad\qquad\qquad\qquad\qquad\qquad$ (A.1.1)
$\qquad$ Else $snowfall_t = 0$ $\qquad\qquad\qquad\qquad\qquad\qquad\qquad\qquad\qquad$ (A.1.2)

Determining if the temperature generates snowmelt, calculating snowmelt and VI.
If $T_t \leq \mathbf{T_M}$

$\qquad$ Then $snowpack_t = snowpack_{t-1} + snowfall_t$ $\qquad\qquad\qquad\qquad$ (A.1.3)
$\qquad$ Else $snowmelt_t = \mathbf{C_M} \times (T_t - \mathbf{T_M}) \times snowpack_{t-1}$ $\qquad\qquad\quad$ (A.1.4)
$\qquad$ And $snowpack_t = snowpack_{t-1} - snowmelt_t$ $\qquad\qquad\qquad\quad$ (A.1.5)
If $T_t > 0$

$\qquad$ Then $VI_t = P_{TOTt} + snowmelt_t$ $\qquad\qquad\qquad\qquad\qquad\qquad$ (A.1.6)
$\qquad$ Else $VI_t = snowmelt_t$ $\qquad\qquad\qquad\qquad\qquad\qquad\qquad\qquad$ (A.1.7)

With:

t = the current daily time step

$T_t$ = the air temperature (°C)

$snowfall_t$ = the snowfall in snow water equivalent (mm)

$P_{TOTt}$ = the total precipitation (mm)

$\mathbf{T_M}$ = the melting temperature (°C)

$snowpack_t$ = the snowpack in snow water equivalent (mm)

$snowpack_{t-1}$ = the snowpack in snow water equivalent at the previous time step (mm)

$snowmelt_t$ = the liquid water produced by snowmelt (mm)

$\mathbf{C_M}$ = the melting coefficient (mm.°C$^{-1}$.d$^{-1}$)

$VI_t$ = vertical inflow (mm)

***Runoff Computation***

Computing the antecedent soil conditions

$$APL_t = \sum_{t=t-\mathbf{t_{API}}}^{t} VI_t \tag{A.1.8}$$

Computing the values of RCN for dry and humid soil conditions based on equations from Monfet (1979)

$$RCN_{dry} = 0.00865 \times \mathbf{f_{runoff}} \times RCN^2 + 0.0145 \times \mathbf{f_{runoff}} \times RCN + 7.39846 \tag{A.1.9}$$

$$RCN_{wet} = -0.00563 \times \mathbf{f_{runoff}} \times RCN^2 + 1.45535 \times \mathbf{f_{runoff}} \times RCN + 10.82878 \tag{A.1.10}$$

Adjusting the RCN value based on the antecedent soil conditions

If $July\ 1^{st} \leq t < September\ 1^{st}$

    If $API_t < 50$

        Then $RCN_t = RCN_{dry}$       (A.1.11)

    If $API_t > 80$

        Then $RCN_t = RCN_{wet}$       (A.1.12)

    Else $RCN_t = \mathbf{f_{runoff}} \times RCN$       (A.1.13)

If $June\ 1^{st} \leq t < July\ 1^{st}$ or $September\ 1^{st} \leq t < October\ 10^{th}$

    If $API_t < 18.5$

        Then $RCN_t = RCN_{dry}$       (A.1.14)

    If $API_t > 37$

        Then $RCN_t = RCN_{wet}$       (A.1.15)

    Else $RCN_t = \mathbf{f_{runoff}} \times RCN$       (A.1.16)

If $October\ 10^{th} \leq t < June\ 1^{st}$

    If $API_t < 11$

        Then $RCN_t = RCN_{dry}$       (A.1.17)

    If $API_t > 22$

        Then $RCN_t = RCN_{wet}$       (A.1.18)

    Else $RCN_t = \mathbf{f_{runoff}} \times RCN$       (A.1.19)

Computing runoff (with condition on the soil frost)

If $\frac{1}{\mathbf{F_T}} \sum_{t=t-\mathbf{F_T}}^{t} T_t > \mathbf{TT_F}$

    Then $R_t = \dfrac{\left[ VI_t - 0.2 \times \left( {1\ 000}/{RCN_t} - 10 \right) \right]^2}{VI_t - 0.8 \times \left( {1\ 000}/{RCN_t} - 10 \right)}$       (A.1.20)

    Else $R_t = VI_t$       (A.1.21)

With:

$API_t$ = the antecedent precipitation index (mm)

$t_{API}$ = the antecedent precipitation index time (d)

RCN = the computed value of runoff curve number for the considered pixel (-)

$f_{runoff}$ = runoff factor (-)

$RCN_{dry}$= the corrected value of runoff curve number for dry soil conditions (for the Quebec environment) (-)

$RCN_{wet}$= the corrected value of runoff curve number for humid soil conditions (for the Quebec environment) (-)

$RCN_t$ = the considered value of runoff curve number for the time step (-)

$F_T$ = the freezing time (d)

$TT_F$ = the threshold temperature for soil frost (°C)

$R_t$ = runoff (mm)

*Lumped soil reservoir*

Computing infiltration as runoff excess
$$Inf_t = VI_t - R_t \tag{A.1.22}$$

Computing saturation excess
If $(\mathbf{sw_m} - sw'_{t-1}) \geq Inf_t$
$\quad\quad$ Then $Excess\ R_t = 0$ $\hfill$ (A.1.23)
$\quad\quad$ Else $Excess\ R_t = Inf - (\mathbf{sw_m} - sw_{t-1}')$ $\hfill$ (A.1.24)
$Total\ R_t = R_t + Excess\ R_t$ $\hfill$ (A.1.25)

Computing the AET based on the soil water content
If $sw_{t-1}' + Inf_t - Excess\ R_t \geq PET_t$
$\quad\quad$ Then $AET_t = PET_t$ $\hfill$ (A.1.26)
$\quad\quad\quad\quad$ $sw_t = sw_{t-1}' + Inf_t - Excess\ R_t - AET_t$ $\hfill$ (A.1.27)
$\quad\quad$ Else $AET_t = sw_t + Inf_t - Excess\ R_t$ $\hfill$ (A.1.28)
$\quad\quad\quad\quad$ $sw_t = 0$ $\hfill$ (A.1.29)

Computing the potential GWR based on the soil water content after the AET computation
If $sw_t > 0$
$\quad\quad$ Then $GWR_t = sw_t \times \mathbf{f_{inf}}$ $\hfill$ (A.1.30)
$\quad\quad\quad\quad$ $sw_t' = sw_t - GWR_t$ $\hfill$ (A.1.31)
$\quad\quad$ Else $GWR_t = 0$ $\hfill$ (A.1.32)
$\quad\quad\quad\quad$ $sw_t' = 0$ $\hfill$ (A.1.33)

With:

$Inf_t$ = infiltration to the soil reservoir (mm)

$\mathbf{sw_m}$ = maximum soil water content in the soil reservoir (mm)

$sw_{t-1}'$ = soil water content at the end of the previous time step (mm)

$Excess\ R_t$ = saturation excess produced by the soil reservoir (mm)

$Total\ R_t$ = total runoff (mm)

$PET_t$ = potential evapotranspiration (mm)

$AET_t$ = actual evapotranspiration (mm)

$sw_t$ = soil water content after the AET computation (mm)

$GWR_t$ = potential GWR (mm)

$\mathbf{f_{inf}}$ = infiltration factor (d$^{-1}$)

$sw_t'$ = soil water content after the AET and GWR computation (mm)

*Model output per grid cell*

$$R_m = \sum_{t=1}^{n} Total\ R_t \tag{A.1.34}$$

$$AET_m = \sum_{t=1}^{n} AET_t \tag{A.1.35}$$

$$GWR_m = \sum_{t=1}^{n} GWR_t \tag{A.1.36}$$

With:

$R_m$ = simulated monthly total runoff (mm)

n = number of days in the considered month

$AET_m$ = simulated monthly AET (mm)

$GWR_m$ = simulated monthly potential GWR (mm)

---

## Author Response (AR3)

Dear Editor,

As suggested in your last review, we have modified our manuscript to include the reference to the model code that is now downloadable here: https://doi.org/10.5683/SP3/EUDV3H

The reference to the model script on Dataverse, "Dubois et al., 2021a", was added L83 and L137 (entire reference hereafter).

In order to advertise for the model script availability, the section 8 Code availability (L593-594) was changed for: "The script of the HydroBudget model is open source and can be downloaded with an application example from: https://doi.org/10.5683/SP3/EUDV3H"

The reference to the model script on Dataverse was added in the section 10 References (L676-677): "Dubois, E., Larocque, M., Gagné, S. and Meyzonnat, G.: HydroBudget – Groundwater recharge model in R. https://doi.org/10.5683/SP3/EUDV3H, 2021a."